# Passenger and freight travel patterns: A cluster analysis based on urban networks

**Soyeong Lee**, **Heesun Joo**\*

Department of Urban Engineering, Gyeongsang National University, Jinju, Republic of Korea

\* hsjoo@gnu.ac.kr

## Abstract

While research on population travel patterns and urban networks has been active, it has primarily focused on passenger travel, leaving freight travel relatively underexplored. This study addresses this gap by analyzing both passenger and freight travel patterns, network structures, and central areas. It uses origin-destination (OD) data, considering total travel volume by purpose and mode. The study applies regular equivalence and power centrality to examine differences in human and logistics flows across South Korea from an urban network theory perspective. The key findings are as follows. First, passenger travel, predominantly short-distance, exhibits lower density and intensity than freight travel. Freight travel, on the other hand, demonstrates strong density across short, medium, and long distances, with more travel routes concentrated around nodal regions. Second, passenger travel forms several polynucleated clusters, including short-distance movements. Conversely, freight travel forms a few extensive clusters that encompass medium and long-distance movements. Third, the spatial interaction of passenger travel is influenced by the OD distance, unlike freight travel. Interestingly, the distance between central areas of freight travel can be longer than that of passenger travel. This may stem from the strategic positioning of certain suburban areas as central areas to optimize logistics efficiency. This study emphasizes the importance of morphological and functional linkages between cities by identifying inter-regional differences in passenger and freight flows. It also proposes spatial planning strategies based on urban hierarchy.

## Introduction

Cities grow and maintain their functions not in isolation but through interactions with neighboring cities. Recognizing the importance of these interactions and relationships, there has been a surge in efforts to analyze the role of nodal regions within networks [1–3]. Network analyses have been conducted that utilize various indicators and examine factors contributing to external effects within urban networks [4–9]. However, most city networks exhibit more morphological than functional trends, highlighting the need for an integrative approach [10]. In South Korea, urban decline and population issues are recognized as significant concerns, prompting research into regional patterns of population migration. Addressing these social problems necessitates a systematic approach that takes into account urban functions and structures [11]. This approach is closely tied to urban network formation, indicating a shift in

**Data availability statement:** The minimal data set required to replicate our study's

findings has been uploaded to Figshare and can be accessed at the following DOI: 10.6084/m9.figshare.26953639. The original data used in this study were provided by the Korea Transport Database (KTDB) and can be accessed at https://www.ktdb.go.kr/.

**Funding:** This work was financially supported by the National Research Foundation of Korea (NRF) (https://www.nrf.re.kr.) via the Ministry of Science and ICT (MSIT) (https://www.msit.go.kr) in the form of a grant (RS-2024-00339015) received by HJ. No additional external funding was received for this study. The funder had no role in study design, data collection and analysis, decision to publish, or preparation of the manuscript.

**Competing interests:** The authors have declared that no competing interests exist.

urban form from "place-centered" to "flow-centered" [1,12]. However, most prior studies on population movement have focused on passenger travel [13–19], with freight travel receiving less attention [20,21], largely due to the lack of reliable data [22]. This study aims to elucidate the patterns, network structures, and regional differences in passenger and freight travel, focusing on their impacts on various aspects of a region's economy, society, and culture. In particular, this study focuses on the technical analysis of the static patterns of passenger and freight travel to elucidate the structural characteristics of urban networks in South Korea. The study primarily employs statistical and graphical methods to analyze passenger and freight travel patterns, while excluding any dynamic analyses related to population decline or regional changes.

The specific objectives of this study are as follows:

1. Examine the current status and movement patterns of passenger and freight travel with the aim of identifying trends in both human and material flows.

2. Identify common sub-regions within passenger and freight travel networks through regular equivalence analysis.

3. Identify primary and secondary central areas and propose strategies for enhancing inter-regional linkages based on the analysis results.

## Related work

### Urban network theory

The urban network theory, which has evolved through various concepts such as "dispersed city," "network city," "city network," "polynucleated metropolitan region," and "polycentric or polynuclear urban region," has been linked to the discourse on sustainable urban policy, particularly in the context of the European Union, since the mid-2000s [1,23–27]. This theory posits that cities emerge from the interactions and mutual development of economic, social, and physical fields. A city is deemed to have the potential for self-sustaining development when the cumulative benefits outweigh the disadvantages over time. An urban network system is defined as a collaborative arrangement where two or more independent cities supplement urban functions through transport and communication infrastructure to achieve economies of scale [27].

Recently, the functional linkage networks between large cities and small to medium-sized cities have garnered increased attention. Such networks encompass three core concepts. First, the principle of mutual cooperation suggests that intercity relationships extend beyond hierarchical structures, fostering economic development through city-to-city collaboration [28]. Second, the concept of network externalities indicates that participation in the network can yield economies of scale and synergy effects. Finally, the network is not predicated on the region's hierarchical structure but emerges from intercity connections sharing similar or dissimilar characteristics through long-distance relations between comparably-sized cities.

Cities offer diverse functions through specialization, with high-level functions potentially provided in lower-tier cities under certain circumstances [28,29]. These network cities, also known as polycentric urban regions, are characterized by the coexistence of multiple specialized cities within a specific area. Polycentric urban areas, similar to network cities, are defined by the presence of polynucleated cities within a certain area and the connectivity among them [30]. However, while two or more cities may coexist within a specific area and exhibit interdependence regarding certain indicators, they may not necessarily share the area's identity or culture. Furthermore, the expansion of network theory has prompted various studies focusing

on the classification of intercity connections, examination of factors causing externalities in network cities, and investigation of the principles governing the organization of intercity networks [4,7,10].

## Prior research on the comparison of passenger travel and freight travel

Spatial interaction, one form of which is intercity movement patterns, is consistently defined as the movement and mobility of objects, ideas, goods, and people between spatially separated parts or places. The movement of people and goods within cities forms a dominant pattern of intercity interactions [31]. Numerous discussions have been held on the relationship between travel flow and urban form since the mid-1980s [32–36]. However, most studies have focused on passenger travel, with less attention given to the interaction between various urban area characteristics and freight travel activities. In conclusion, urban networks are constructed not only by human flows but also by material flows that underpin national or regional industries. Therefore, understanding urban networks solely through daily passenger travel is insufficient [37].

Passenger and freight movements are fundamentally different. The former, defined as passenger flow, involves the movement of individuals for various reasons, while the latter, known as freight flow, is solely concerned with transporting goods from one point in the supply chain to another [38]. Analysis of population movement data is typically used to understand the morphological connections between cities, while freight travel data is analyzed to identify functional linkages [7]. As depicted in Fig 1, passenger movement is generally associated with short distances, constrained by the distance/time ratio. In contrast, freight movement encompasses a wider area, closely aligned with the principle of comparative advantage [39]. This distinction underscores the clear difference in characteristics between human mobility and logistics flows [18,40–43].

Intercity passenger and freight flows do not always align [39,43,44]. While passengers may undertake multiple journeys within an urban area daily, amounting to thousands annually, goods are primarily transported from outside the city to the urban area. Following this, they are typically moved to one or two additional locations within the region before

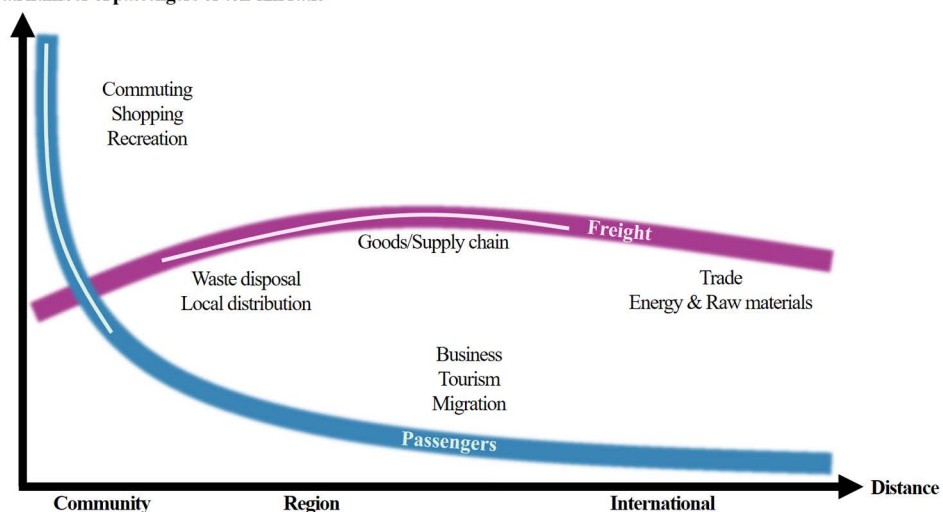

**Fig 1. Passenger and freight mobility.** Source: Rodrigue JP. The geography of transport systems. 5th ed. Routledge; 2020.

consumption. Consequently, the spatial interactions of freight movement are minimally affected by the OD distance. The dominant position and network structure within a region can vary, depending on the unique characteristics of each city [37,45]. It has been observed that when travel pattern-related flows are used as a measure of functional connectivity, they form a hierarchical structure [46]. In summary, while most passenger travel tends to gravitate towards nearby destinations [47–49], freight movement is more closely associated with specific suburban areas.

## Research on network structures and clusters

Research on interregional travel patterns has long been a focus in urban planning [50]. However, previous conclusions were deemed limited due to the lack of large-scale data and appropriate methodologies [51]. The advent of network theory and sensor technology has addressed these limitations, enabling the use of extensive data in various studies to analyze urban systems. These studies have examined diverse areas such as firms and global cities [52–56], finance [19,57,58], transportation, including taxis, railways, and aviation [59–63], cellular base stations and call logs [64–68], population movement and household surveys [69–71], and public transport smart cards [72–76].

Studies employing traditional network indicators can be categorized into three groups: those examining network structure characteristics, spatial structural changes, and network efficiency. Regarding network structure characteristics, social network analysis has been used to study the centrality and urban system of global cities, based on the location information of the headquarters and branches of the world's top 500 multinational corporations [56]. Similarly, Thiemann et al. [57] analyzed the dominance of each city within the global urban system using air passenger data. Liu et al [60] used taxi passenger data in Shanghai to analyze intra-city movement patterns and urban structure, identifying the city's sub-regional structure and characteristics. Saberi et al. [70] compared the network characteristics of passenger demand patterns in Chicago and Melbourne, finding similar characteristics despite differences in topography and urban structure. Lee et al. [64] highlighted the importance of establishing regional living zones by examining movement patterns and structural characteristics of the working population using mobile base station data.

In the spatial structural change context, Alderson et al. [55] analyzed the reconstitution and impact of the global urban system using multinational corporation location data, but failed to provide evidence of increased inequalities between cities. A study analyzing global air passenger flows and cross-border relations suggested that the influence between cities worldwide is somewhat integrated due to the growth of major cities in developing countries and emerging economies [54]. Zhong et al. [75] used smart card data to measure mobility diversity and volatility, finding that this volatility appears at various levels. Guimerà et al. [62] analyzed the structure of the global air transport network, revealing that the structure of various communities within the network, in addition to geographical factors, is critical to predicting centrality. Iqbal et al. [68] analyzed highway travel networks to reveal the correlation between regional economic development and travel patterns, observing changes in spatial interactions between centrality and GDP.

Regarding network efficiency, Salisbury and Barnett [58] identified the centrality of the interregional financial network using fund transfer data between credit cards and banks. Huang et al. [59] reviewed the effect of urban network externalities on urban growth empirically, using train operation frequency data, and determined which city could receive more benefits. Boyd et al. [52] critically analyzed the study by Neal [61], which proposed a new measure of recursive power and recursive centrality in the global urban network, concluding that the existing measure of eigenvector centrality is more useful. Wall and van der Knaap [53]

analyzed the network of the top 100 multinational corporations and subsidiaries in 2,259 cities worldwide, demonstrating how cities are interconnected in various industries and services, and how these connections operate in the economic system.

Urban network theory provides insights into the connections between cities by region, offering theoretical and policy alternatives to address developmental imbalances at both national and regional levels. It serves as an analytical system that can examine how cities are interconnected and how these connections influence various societal aspects, including the economy. However, the theory of network cities has been criticized for its lack of consideration for the real local community economy [77]. It requires systematization of analytical techniques to empirically examine the connectivity of network cities, expansion of connectivity indicators, reconfirmation of the normative nature inherent in network cities, the establishment of cooperative governance for network cities, and exploration of ways to develop internal urban communities linked to network cities. Even so, the emphasis has shifted toward the functional connectivity of cities rather than their geographical proximity, yielding many studies that apply the concept of network cities and offer theoretical justifications for the effects of network cities as well as empirical research conducted in various aspects.

A review of the literature reveals that most studies have explored complex social phenomena related to urban planning, which is traditionally difficult to access, by actively using large-scale data and network analysis methodologies. These studies recognize the importance of network structures and focus on revealing how intercity connectivity and structural characteristics affect social and economic dynamics, such as regional economic development, travel patterns, and population movement. Studies focusing on network structure characteristics emphasize the analysis of individual cities or structural locations, such as urban centrality, dominance, and sub-regional structure. Studies related to spatial structural changes of networks primarily address structural changes and redistribution of influence that occurs within urban systems over time. Studies on network efficiency differ in that they analyze the flow and efficiency of information, resources, and logistics within the network and focus on empirically reviewing how the network can be optimized and which places are occupied by certain cities within the network.

This study distinguishes itself from others in several ways. First, regarding scope, it compares the morphological and functional network structures using passenger and freight travel data of a broad area, including 17 si/do (cities/provinces) and 250 si/gun/gu (cities/counties/districts) in South Korea. Second, for multifaceted data analysis, this study uses total travel volume, encompassing various purposes of passenger and freight travel and data by tonnage in its analysis, focusing on specific travel purposes or modes. Third, regarding methodology, regional clustering was performed using REGGE algorithms, and intercity connectivity, hierarchy, and centrality were quantitatively analyzed using power centrality indicators. In summary, this study stands out from previous ones as it comprehensively analyzes the connectivity, regular equivalence, interregional central area, and role between nodal regions of the urban network using a wide spatial scope and passenger and freight travel data of the entire South Korea.

## Data and methods

### Data

This study encompasses all of South Korea, analyzing 17 si/do and 250 si/gun/gu. The 2019 data includes the most recent OD data for passenger and freight travel. The Korea Transport Database (https://www.ktdb.go.kr/) provided the data, which, for passenger travel, comprised OD data on total travel volume, including seven travel purposes such as commuting,

schooling, working, shopping, home returning, leisure/entertainment/visiting relatives, and others. This data was derived from the total travel volume of linked trips between si/gun/gu. For freight travel, the data included OD data of total travel volume, covering light, medium, and heavy tonnage classes, using the total travel volume of unlinked truck trips between si/gun/gu.

The study's objective is to compare the patterns, distance, hierarchical structure, and centrality of interregional human and material networks, necessitating interregional flow data. Therefore, this study utilized the travel OD of 17 si/do and 250 si/gun/gu in South Korea, based on the passenger and freight OD travel volume in 2019. The value was processed as 0 when i = j, using Netminer for analysis. Passenger and freight flows are crucial in defining power centrality. Analyzing travel volume relative to population at nodal points reveals that centrality may change with alterations in administrative districts. Furthermore, differences in trip departure and arrival volume by node mean that the ratio of arrival to departure varies at each point, affecting the centrality of the relevant point. However, the actual effect on centrality can be accurately reflected by calculating and applying the average of total trip departure and arrival volume. To adjust these differences, travel volume was corrected by dividing it by the average population of each region. In the analysis of passenger and freight travel, the number of passengers and vehicles is more significant than the presence of travel, so it was converted to a symmetric matrix before the analysis. Since the number of passengers and vehicles is small compared to the population, the number of passengers per 1,000 was used as the flow rate, as shown in Equation (1).

$$R_{ij} = R_{ji} = \frac{\left(X_{ij} + X_{ji}\right)}{2} / \frac{\left(P_i + P_j\right)}{2} \times 1,000. \tag{1}$$

This study used threshold settings to construct an OD matrix for analyzing interregional passenger and freight travel. An alternative method involves calculating the ratio of trip inflow (outflow) for each origin (destination) to the total trip inflow (outflow) for each destination (origin) and extracting only the connection lines where the ratio is 5% or more [78]. However, this method has a drawback: the travel volume, which accounts for 5%, varies greatly depending on the size of the total trip inflow (outflow) at each nodal point. Therefore, this study established the number of connection lines by setting an absolute threshold after excluding the internal travel volume of each nodal point. The thresholds were set at 1,000 trips for passenger travel and 100 trips for freight travel. If the travel volume between two nodal points is less than 1,000 trips or 100 trips, the value is set to '0'; for values above the threshold, a phase structure was built using Equation (1). This method allows for the construction of a phase structure that accurately reflects the actual flow between nodal points where at least a certain level of interaction occurs.

**Methods.** The research methodology is divided into three sections. The first involves identifying travel patterns to ascertain the strength of intercity flows by measuring the connectivity between nodal regions within passenger and freight networks. The second step involves identifying the regular equivalence of urban networks to analyze the similarity of positions or characteristics among 17 si/do and group them accordingly. This step utilizes REGGE algorithms to measure the similarity between nodal regions. The third step involves deriving primary and secondary central areas to highlight the significance and roles of regions, as defined by the proportion, function, and interregional relationships determined by the interregional passenger and freight travel network patterns. This step measures the importance of nodal regions within the network using power centrality indices for each clustered region. Chapter 2 of this study explores urban network theory and reviews various

practical studies related to network and cluster analysis. Chapter 3 outlines the methodologies for empirical analysis, while Chapter 4 presents the results of cluster and centrality analysis, including the current status. Chapter 5 provides implications (see Fig 2).

**Regular equivalence (REGGE).** Location analysis in social network analysis aims to identify actors connected in similar patterns within the network (i.e., individuals or regions exhibiting similar relationship forms) and classify them into comparable groups [79]. Consequently, regions in the same location play similar roles or hold similar positions. In the context of travel networks, a region's position reflects its connectivity within the country. The concept of regular equivalence is applied here to measure the network position of regions [80,81]. This concept mirrors the structure of relationships between different regions based on the similarity of the positions and roles performed by regions within the network, making regular equivalence a crucial measure for understanding intercity roles and positions in network analysis [82].

Levels of regular equivalence are measured using REGGE algorithms [82,83]. These algorithms estimate the similarities between two specific regions when analyzing interregional relationships. The process begins with an initial estimate of the similarities in interregional travel. Then, estimates of similarity between the two regions are adjusted by examining how similar the two regions are to other regions to which they are each connected. By repeating this process multiple times, quantitative figures on how similar and interchangeable the two regions are to each other are obtained. The scale of regular equivalence generated by REGGE algorithms is measured by Equation (2) [82–84].

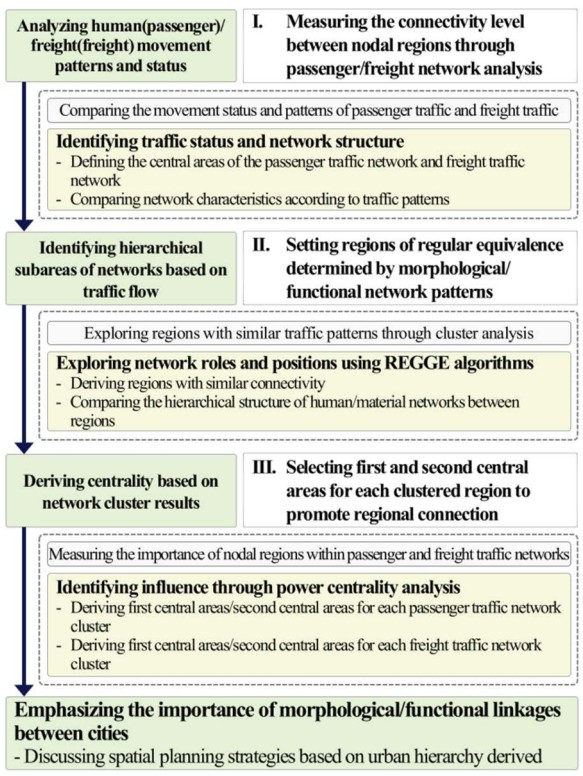

**Fig 2. Research flow.**

$$M_{ij}^{t+1} = \frac{\sum_{k=1}^{g} \max_{m=1}^{g} \sum_{r=1}^{R} (_{ijr} M_{kmr}^{t} + _{jir} M_{kmr}^{t})}{\sum_{k=1}^{g} \max_{m} * \sum_{r=1}^{R} (_{ijr} Max_{kmr} + _{jir} Max_{kmr})}, \tag{2}$$

where $M_{ij}^{t+1}$ denotes the regular equivalence between regions i and j at the (t + 1)th iteration, based on the travel network. The denominator signifies the maximum possible consistency when the connections of region i to all other regions (k) perfectly align with the connections of region j to all other regions (m). Here, the changing entities for regions i and j (k and m) exhibit regular similarity. The numerator optimally aligns the relationship of region j with m to that of region i with k, a process weighted by the regular equivalence of k and m from the previous iteration. Consequently, this algorithm identifies the optimal match in the connections between region i and all other regions, and between region j and all other regions. These connections are weighted according to their similarity with other regions in the network and are calculated by the maximum theoretically possible consistency [85]. Thus, $M_{ij}^{t+1}$, the regular equivalence, represents the inverse function of the degree to which the connections of region i with other regions align with the connections of region j with other regions. The similarity of interregional travel flows is reassessed after each iteration [85], and the degree of similarity between the regions in the network is quantified. The resulting similarity ranges from 0 to 1, where 0 signifies complete nonequivalence between two regions, and 1 indicates perfect regular equivalence.

## Network centrality analysis (power centrality)

Centrality in a network signifies an individual's influence and power within that network. Analyzing centrality allows for the identification of key actors within the network, or those receiving significant attention. These influential actors are often referred to as central or hub nodes. The primary centrality indices in a network include degree centrality, closeness centrality, and betweenness centrality [86]. Other indices, such as eigenvector centrality [87], page rank [88], and structural holes [89], also exist. Regardless of the index used, centrality is not an absolute measure; instead, it indicates relative ranking. Therefore, normalized centrality values may be employed, depending on the network's scale (e.g., size and density). Power centrality, a new centrality index, was developed to address the variability in results produced by different centrality indices [90]. Also known as Bonacich beta centrality, it supplements degree and eigenvector centrality. In this context, β (beta) represents the weight needed to calculate Bonacich's centrality index. A small β value emphasizes the local structure surrounding specific actors, while a large β value considers the entire network structure.

The centrality of a region can be analyzed by treating the origin and destination as nodes in the network, representing interactions between nodes as links, and examining the movement flow within the network [91,92]. High centrality in adjacent regions suggests the potential activation of economic and cultural activities in the region. Interregional interactions are often confrontational, leading to a negative β value. However, previous research set β as 0 (ignoring the impact of indirect connections requiring multiple steps in degree centrality) or 1/(λ max) (maximizing the impact of indirect connections requiring multiple steps in eigenvector centrality, where λ is the maximum eigenvalue). For travel networks with confrontational interactions, these centrality indices may sometimes yield inconsistent results. Therefore, power centrality can be calculated as shown in Equation (3).

$$c_i(\alpha, \beta) = \sum_{j}(\alpha + \beta c_j)R_{ij}, \tag{3}$$

where $c_i$ is a vector containing the beta centrality of nodes as elements, α is the range used to standardize centrality values, β represents the range of weights based on the distance with nodes, and $R_{ij}$ is an element of the adjacency matrix. The β value, as described above, assigns more weight to the network structure surrounding a specific high-centrality point when calculating power centrality. This suggests that larger β values place more emphasis on a broader network structure. Consequently, the β value cannot be uniformly determined when nodal points represent individual regions. By setting β=−1/(λ max), assuming confrontational interregional interaction, it is possible to relatively reduce the impact of the distant network structure compared to β=1/(λ max), as shown in Equation (4).

$$C_{\beta} = \alpha \sum_{k=0}^{\infty} \beta^k R^{k+1} = \alpha\left(R1 - \beta R^2 1 + \beta^2 R^3 1 - \beta^3 R^4 1 \cdots\right). \tag{4}$$

## Results

### Travel status

As depicted in Table 1 and Fig 3, South Korea's passenger travel in 2019 comprised 13,849,020 inter-si/do trips and 45,963,003 inter-si/gun/gu trips daily. The capital regions (Seoul Special City, Incheon Metropolitan City, Gyeonggi-do) accounted for at least 50% of the total travel volume, with most travel concentrated in this region. Incheon Metropolitan City and Gyeonggi-do have more arrival volume than departure volume, while Seoul Special City has more departure volume than arrival volume. This suggests that Seoul, a hub for business, education, and cultural activities, draws people from its satellite cities, Incheon and Gyeonggi-do. The high trip departure volume in Seoul indicates that many individuals residing in Incheon or Gyeonggi-do commute to Seoul for work or school. When analyzed by si/gun/gu,

**Table 1. Passenger travel status (Trip/day, %).**

| Classification | Movements between si/do | | | | Movements between si/gun/gu | | | |
|---|---|---|---|---|---|---|---|---|
| | Departure volume | | Arrival volume | | Departure volume | | Arrival volume | |
| Seoul | 3,846,725 | 28% | 3,760,801 | 27% | 13,029,681 | 28% | 12,943,757 | 28% |
| Busan | 547,385 | 4% | 539,030 | 4% | 4,026,211 | 9% | 4,017,857 | 9% |
| Daegu | 507,775 | 4% | 508,036 | 4% | 2,832,127 | 6% | 2,832,388 | 6% |
| Incheon | 1,046,503 | 8% | 1,119,696 | 8% | 2,663,049 | 6% | 2,736,242 | 6% |
| Gwangju | 301,996 | 2% | 296,925 | 2% | 1,728,014 | 4% | 1,722,943 | 4% |
| Daejeon | 406,773 | 3% | 384,253 | 3% | 1,624,349 | 4% | 1,601,829 | 3% |
| Ulsan | 219,291 | 2% | 202,596 | 1% | 1,206,679 | 3% | 1,189,985 | 3% |
| Gyeonggi | 3,994,031 | 29% | 4,019,733 | 29% | 9,931,947 | 22% | 9,957,649 | 22% |
| Gangwon | 207,832 | 2% | 210,812 | 2% | 623,109 | 1% | 626,089 | 1% |
| Chungbuk | 350,926 | 3% | 341,639 | 2% | 1,210,940 | 3% | 1,201,653 | 3% |
| Chungnam | 496,964 | 4% | 492,487 | 4% | 1,359,585 | 3% | 1,355,108 | 3% |
| Jeonbuk | 168,907 | 1% | 167,538 | 1% | 1,068,468 | 2% | 1,067,100 | 2% |
| Jeonnam | 328,891 | 2% | 332,072 | 2% | 924,370 | 2% | 927,552 | 2% |
| Gyeongbuk | 600,712 | 4% | 611,529 | 4% | 1,365,236 | 3% | 1,376,053 | 3% |
| Gyeongnam | 577,136 | 4% | 594,407 | 4% | 1,923,938 | 4% | 1,941,208 | 4% |
| Jeju | 40,119 | 0% | 40,792 | 0% | 238,243 | 1% | 238,915 | 1% |
| Sejong | 207,055 | 1% | 226,674 | 2% | 207,055 | 0% | 226,674 | 0% |
| Total | 13,849,020 | 100% | 13,849,020 | 100% | 45,963,003 | 100% | 45,963,003 | 100% |

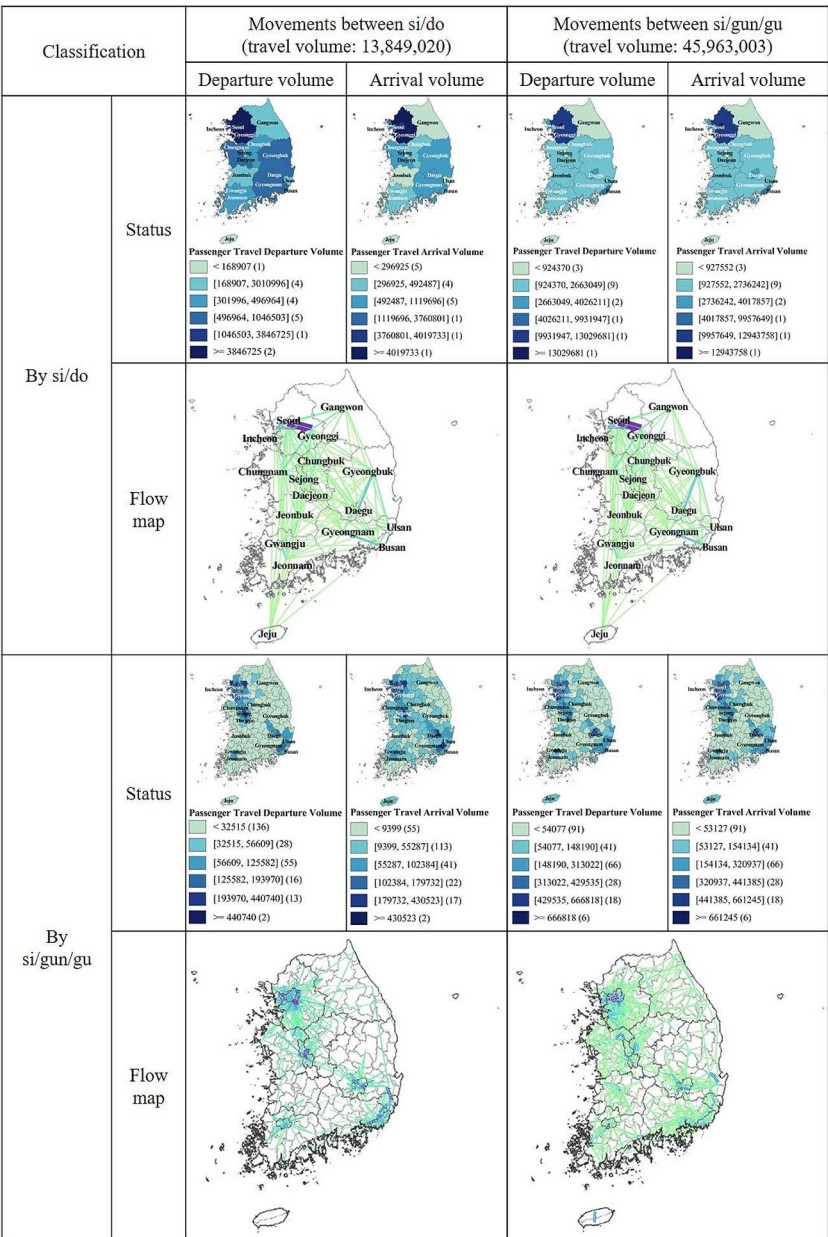

Note: A method exists for representing at least 5% of travel when forming connection lines in the travel network. However, the travel volume, which constitutes this 5%, can greatly vary depending on the total trip inflow (outflow) at each nodal point. Therefore, this study constructed a flow map by excluding the internal travel volume at each nodal point and setting an absolute threshold of 1,000 trips.

**Fig 3. Passenger travel flows and patterns.**

the travel patterns were similar to those at the si/do level, but with a greater concentration in metropolitan cities (Busan Metropolitan City, Daegu Metropolitan City) and the Seoul Capital Area. Gyeongsangnam-do, despite being adjacent to three metropolitan cities (Busan, Daegu, and Ulsan), had higher travel volume only in certain regions near these cities.

Freight travel, as shown in Table 2 and Fig 4, was 1,052,474 inter-si/do vehicles and 2,259,505 inter-si/gun/gu vehicles daily in 2019. The capital regions accounted for at least 45%

**Table 2. Freight travel status (Vehicle/day, %).**

| Classification | Movements between si/do | | | | Movements between si/gun/gu | | | |
|---|---|---|---|---|---|---|---|---|
| | Departure volume | | Arrival volume | | Departure volume | | Arrival volume | |
| Seoul | 126,366 | 12% | 123,670 | 12% | 318,418 | 14% | 315,722 | 14% |
| Busan | 67,538 | 6% | 68,427 | 7% | 159,074 | 7% | 159,962 | 7% |
| Daegu | 30,707 | 3% | 32,733 | 3% | 94,447 | 4% | 96,472 | 4% |
| Incheon | 98,779 | 9% | 95,486 | 9% | 186,170 | 8% | 182,877 | 8% |
| Gwangju | 20,912 | 2% | 21,360 | 2% | 49,940 | 2% | 50,388 | 2% |
| Daejeon | 19,189 | 2% | 19,150 | 2% | 55,747 | 2% | 55,709 | 2% |
| Ulsan | 25,944 | 2% | 27,576 | 3% | 51,764 | 2% | 53,397 | 2% |
| Gyeonggi | 249,235 | 24% | 262,007 | 25% | 597,146 | 26% | 609,918 | 27% |
| Gangwon | 40,807 | 4% | 37,768 | 4% | 70,408 | 3% | 67,369 | 3% |
| Chungbuk | 59,075 | 6% | 60,152 | 6% | 91,758 | 4% | 92,836 | 4% |
| Chungnam | 75,000 | 7% | 73,439 | 7% | 118,480 | 5% | 116,919 | 5% |
| Jeonbuk | 38,678 | 4% | 36,005 | 3% | 68,445 | 3% | 65,771 | 3% |
| Jeonnam | 45,031 | 4% | 43,186 | 4% | 89,492 | 4% | 87,647 | 4% |
| Gyeongbuk | 64,930 | 6% | 62,722 | 6% | 120,625 | 5% | 118,418 | 5% |
| Gyeongnam | 80,238 | 8% | 79,058 | 8% | 163,543 | 7% | 162,364 | 7% |
| Jeju | 0 | 0% | 0 | 0% | 14,004 | 1% | 14,004 | 1% |
| Sejong | 10,045 | 1% | 9,733 | 1% | 10,045 | 0% | 9,733 | 0% |
| Total | 1,052,474 | 100% | 1,052,474 | 100% | 2,259,505 | 100% | 2,259,505 | 100% |

of the total freight travel volume. Gyeonggi-do receives more freight travel than it sends out, suggesting that it serves as a consumption point for manufacturing businesses, logistics, and distribution centers, or as a hub for storage and distribution. This implies that goods produced in Seoul and Incheon are likely imported by firms in Gyeonggi-do or redistributed to other regions through Gyeonggi-do. In contrast, Seoul Special City and Incheon Metropolitan City are production, export, or departure points as they have more freight travel departures than arrivals. This is likely due to import and export activities through Incheon's international airport and port, as well as various commercial and industrial activities in Seoul. The travel volume gap between the Seoul Capital Area and non-capital areas such as Gyeongsangnam-do, Chungcheongnam-do, Busan Metropolitan City, Gyeongsangbuk-do, and Chungcheongbuk-do was narrower for freight than for passenger travel, revealing different patterns in human and logistics networks. Finally, the si/gun/gu analysis yielded similar results to the si/do level. The flow map of si/gun/gu showed a more even exchange of influence with surrounding areas than passenger travel, indicating a functional trade-off.

In summary, travel volume significantly differs across regions. Seoul, Daejeon, Chungnam, and Jeonbuk, characterized by higher departure volume than arrival volume in both passenger and freight travel, serve as economic and social hubs. These regions are home to diverse industries, including technology, manufacturing, and agriculture, leading to the active production and distribution of goods and services. The high departure volume of freight in these regions suggests a vibrant logistics sector. Conversely, Daegu and Gyeonggi, where arrivals surpass departures in both passenger and freight travel, function as centers of consumption and distribution. Therefore, while Seoul, Daejeon, Chungnam, and Jeonbuk are primary producers of goods and services, Daegu and Gyeonggi are major consumers, illustrating the complementary economic roles of these regions.

Regions with high passenger travel but low freight departures, such as logistics, manufacturing, or agriculture, may be hubs of human resources, indicating fewer industries that

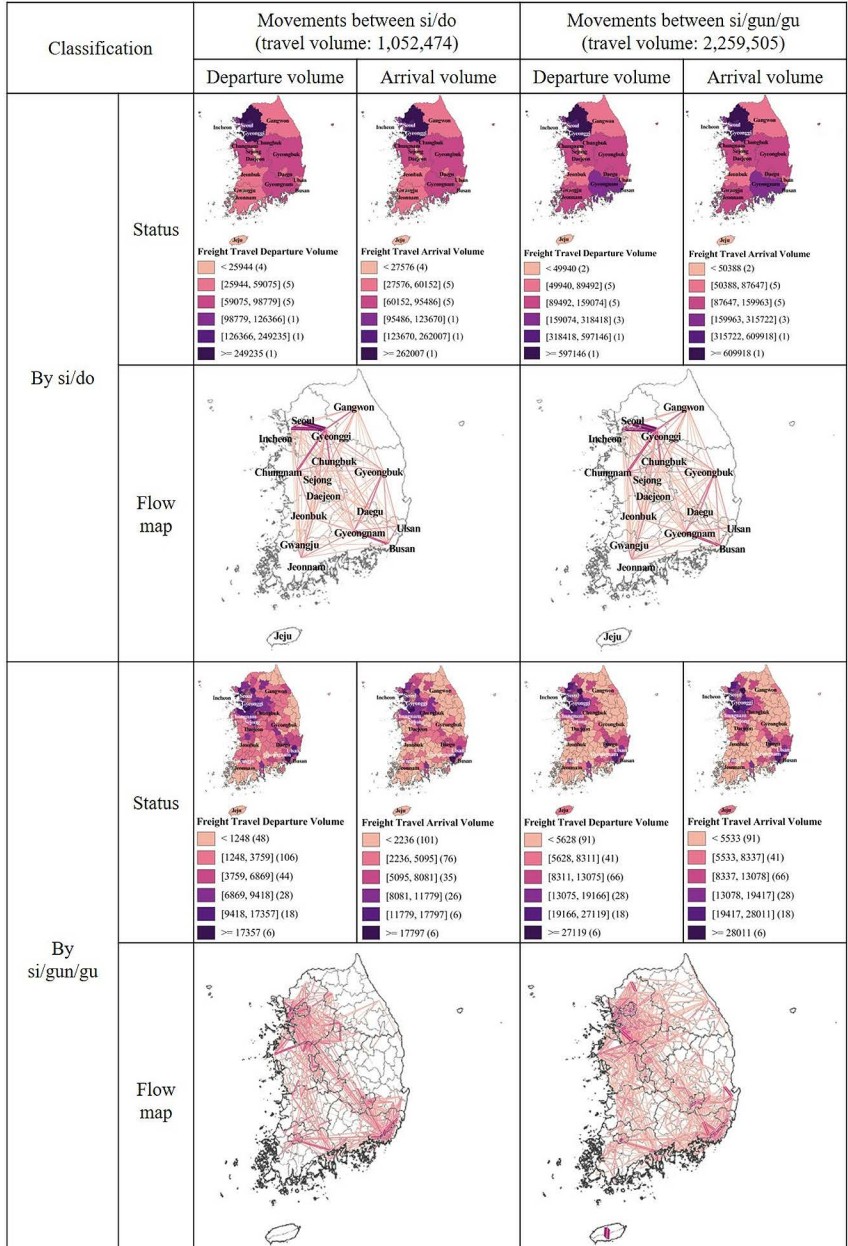

**Fig 4. Freight travel flows and patterns.**

generate significant freight compared to other regions. This highlights the diversity of each region's economic role and industrial structure. Regions like Incheon, Gangwon, Jeonnam, Gyeongbuk, Gyeongnam, Jeju, and Sejong, except for Incheon, generally have a low population density and are known for manufacturing, agriculture, fishing, and specialized industries. These regions have higher departure volume than the arrival volume for freight travel,

suggesting active trade or distribution, with goods or resources produced within the region being moved to other regions.

Incheon, home to a major international trade port, plays a crucial role in collecting and redistributing goods domestically and internationally. In contrast, Busan, Gwangju, Ulsan, and Chungbuk, characterized by high population density and economic activities centered on commerce, services, and light industries, may attribute their high passenger travel departure volume to these activities. A comparison of passenger and freight travel patterns reveals that passenger travel patterns have a lighter intercity travel density than freight travel patterns and exhibit balanced and even patterns, resembling a distributed connected set. The extensive interactions in these regions may be attributed to the human network's close relation to travel routes, such as subways, express bus terminals, train stations, and airports, in addition to cars. Freight travel patterns, however, display greater density and intensity than passenger travel patterns, with more long-distance movements observed due to the clear purpose of goods movement.

This interpretation closely aligns with the composition of South Korea's gross domestic product (GDP). Approximately 60% of South Korea's GDP is generated by the services sector, while around 30% is derived from manufacturing. This economic structure can result in variations in passenger and freight transportation between cities, depending on their economic activities and industrial structure. For instance, a city like Seoul, which serves as the administrative hub for major corporations, is likely to experience an increase in passenger travel related to the service sector. In contrast, a city like Incheon, which hosts a major international port, is likely to experience more active freight transport due to its role as a hub for imports and exports.

## Travel network equivalence analysis

Network analysis offers intriguing insights into the sub-structures within a network, serving as a structural foundation for stratification. Passenger and freight travel networks, for instance, comprise numerous interconnected subnetworks or travel blocks. These blocks are more substantially linked to regions within the same network than to those in other network blocks. Travel network analysis reveals core regions with tight connections and peripheral regions with looser connections. This study aims to explore the sub-structures of regions in each block by categorizing the core and peripheral regions between the origin and destination within the travel network. The X and Y coordinates represent the eigenvalues derived from the clustering matrix, with regions within the shortest average network distance included in a single cluster. The regular equivalence analysis of the passenger travel network (Fig 5) reveals five clusters from eight groups for inter-si/do movements. These clusters are located in Quadrant 4 (Cluster 1) with high trip departure and arrival volume, Quadrant 2 (Cluster 1) with low trip departure and arrival volume, Quadrant 3 (Cluster 3) with high trip departure volume but low arrival volume, and Quadrant 1 (Clusters 4 and 5) with high trip arrival volume but low departure volume. For movements between si/gun/gu, six clusters were formed from eight groups, located in Quadrant 4 (Clusters 2, 3, and 6) with high trip departure and arrival volume, Quadrant 3 (Cluster 1) with high trip departure volume but low arrival volume, and Quadrant 1 (Clusters 4 and 5) with high trip arrival volume but low departure volume.

The regular equivalence analysis of the freight travel network (Fig 6) shows five clusters from six groups for inter-si/do movements. These clusters are located in Quadrant 4 (Clusters 1 and 3) with high trip departure and arrival volume, Quadrant 2 (Cluster 5) with low trip departure and arrival volume, Quadrant 3 (Cluster 2) with high trip departure volume but low arrival volume, and Quadrant 1 (Cluster 4) with high trip arrival volume but low departure volume. For movements between si/gun/gu, four clusters were formed from six

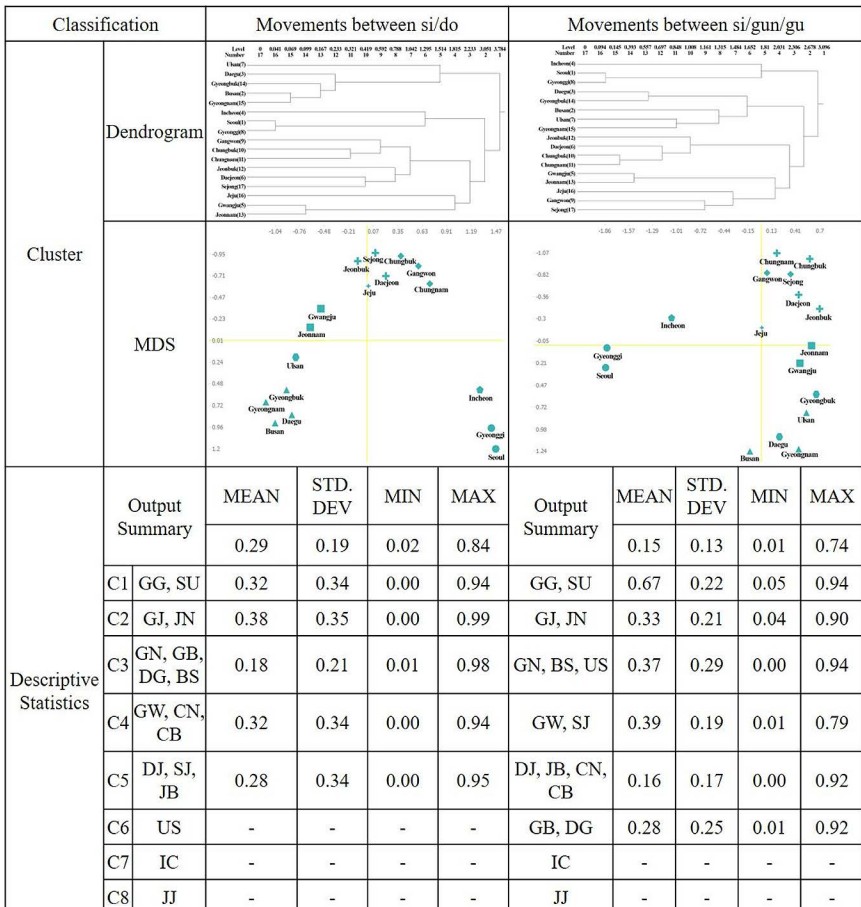

| Classification | | Movements between si/do | | | | Movements between si/gun/gu | | | |
|---|---|---|---|---|---|---|---|---|---|
| Descriptive Statistics | Output Summary | MEAN | STD. DEV | MIN | MAX | Output Summary | MEAN | STD. DEV | MIN | MAX |
| | | 0.29 | 0.19 | 0.02 | 0.84 | | 0.15 | 0.13 | 0.01 | 0.74 |
| | C1 GG, SU | 0.32 | 0.34 | 0.00 | 0.94 | GG, SU | 0.67 | 0.22 | 0.05 | 0.94 |
| | C2 GJ, JN | 0.38 | 0.35 | 0.00 | 0.99 | GJ, JN | 0.33 | 0.21 | 0.04 | 0.90 |
| | C3 GN, GB, DG, BS | 0.18 | 0.21 | 0.01 | 0.98 | GN, BS, US | 0.37 | 0.29 | 0.00 | 0.94 |
| | C4 GW, CN, CB | 0.32 | 0.34 | 0.00 | 0.94 | GW, SJ | 0.39 | 0.19 | 0.01 | 0.79 |
| | C5 DJ, SJ, JB | 0.28 | 0.34 | 0.00 | 0.95 | DJ, JB, CN, CB | 0.16 | 0.17 | 0.00 | 0.92 |
| | C6 US | - | - | - | - | GB, DG | 0.28 | 0.25 | 0.01 | 0.92 |
| | C7 IC | - | - | - | - | IC | - | - | - | - |
| | C8 JJ | - | - | - | - | JJ | - | - | - | - |

\* BS: Busan, CB: Chungbuk, CN: Chungnam, DG: Daegu, DJ: Daejeon, GB: Gyeongbuk, GG: Gyeonggi, GJ: Gwangju, GN: Gyeongnam, GW: Gangwon, IC: Incheon, JB: Jeonbuk, JJ: Jeju, JN: Jeonnam, SJ: Sejong, SU: Seoul, US: Ulsan

**Fig 5. Passenger REGGE cluster dendrogram and multidimensional scaling results.**

groups, located in Quadrant 4 (Clusters 3 and 4) with high trip departure and arrival volume, Quadrant 3 (Cluster 1) with high trip departure volume but low arrival volume, and Quadrant 1 (Cluster 2) with high trip arrival volume but low departure volume. The passenger travel network analysis results show clear quadrant clustering, grouping morphologically similar regions. However, the freight travel network analysis results differ, grouping similar regions even across different quadrants, indicating functional connections.

## Centrality analysis results

Power centrality was calculated to measure regional centrality, employing interregional passenger and freight flows (Equation 1). The initial results, presented in Fig 7, identify the primary and secondary central areas for both passenger and freight travel among si/do. Chungnam and Sejong emerged as the regions where both types of travel shared the same primary and secondary central areas. Gwangju and Gyeongnam, while also sharing central areas, differed in their order of precedence. In some regions, only one central area coincided with both types of travel. For instance, Daejeon and Jeonbuk shared the same primary central area for both passenger and freight travel. Similarly, Gyeongbuk, Chungbuk,

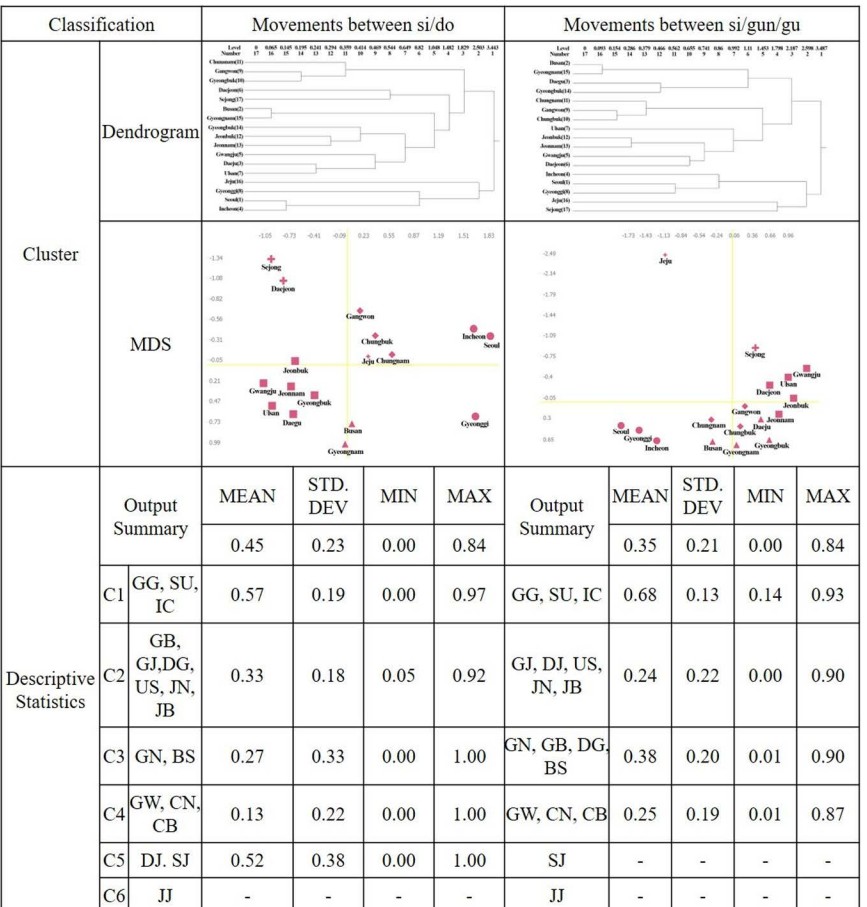

| Classification | | Movements between si/do | | | | Movements between si/gun/gu | | | |
|---|---|---|---|---|---|---|---|---|---|
| Dendrogram | | | | | | | | | |
| Cluster | MDS | | | | | | | | |
| Descriptive Statistics | Output Summary | MEAN | STD. DEV | MIN | MAX | Output Summary | MEAN | STD. DEV | MIN | MAX |
| | | 0.45 | 0.23 | 0.00 | 0.84 | | 0.35 | 0.21 | 0.00 | 0.84 |
| | C1 | GG, SU, IC | | | | GG, SU, IC | | | | |
| | | 0.57 | 0.19 | 0.00 | 0.97 | | 0.68 | 0.13 | 0.14 | 0.93 |
| | C2 | GB, GJ,DG, US, JN, JB | | | | GJ, DJ, US, JN, JB | | | | |
| | | 0.33 | 0.18 | 0.05 | 0.92 | | 0.24 | 0.22 | 0.00 | 0.90 |
| | C3 | GN, BS | | | | GN, GB, DG, BS | | | | |
| | | 0.27 | 0.33 | 0.00 | 1.00 | | 0.38 | 0.20 | 0.01 | 0.90 |
| | C4 | GW, CN, CB | | | | GW, CN, CB | | | | |
| | | 0.13 | 0.22 | 0.00 | 1.00 | | 0.25 | 0.19 | 0.01 | 0.87 |
| | C5 | DJ. SJ | 0.52 | 0.38 | 0.00 | 1.00 | SJ | - | - | - | - |
| | C6 | JJ | - | - | - | - | JJ | - | - | - | - |

\* BS: Busan, CB: Chungbuk, CN: Chungnam, DG: Daegu, DJ: Daejeon, GB: Gyeongbuk, GG: Gyeonggi, GJ: Gwangju, GN: Gyeongnam, GW: Gangwon, IC: Incheon, JB: Jeonbuk, JJ: Jeju, JN: Jeonnam, SJ: Sejong, SU: Seoul, US: Ulsan

**Fig 6. Freight REGGE cluster dendrogram and multidimensional scaling results.**

and Gangwon had the same primary central area for passenger travel and secondary for freight travel.

Moreover, Gyeonggi and Daegu shared the secondary central area for passenger travel and the primary for freight travel. However, no regions were observed where the secondary central areas for both types of travel coincided. Despite these similarities, Seoul, Jeonnam, and Busan exhibited different central areas, and no comparison targets were available for Incheon and Ulsan. Specifically, the differences in the influence and ranking of primary and secondary central areas in passenger and freight travel networks across various cities and provinces can be analyzed. For passenger traffic, the primary central area with the highest influence is Gyeongsan-si in Gyeongbuk (C3), scoring 4.610, followed by Suseong-gu in Daegu (C3) with a score of 3.357, Gangnam-gu in Seoul (C2) with 3.221, Bundang-gu in Seongnam-si, Gyeonggi (C2) with 3.093, Naju-si in Jeonnam (C2) with 2.196, and Buk-gu in Gwangju (C2) with 2.033. For freight travel, Bucheon-si in Gyeonggi (C1) tops the list with a score of 4.354, followed by Gyeongsan-si in Gyeongbuk (C2) with 3.716, Bupyeong-gu in Incheon (C1) with 3.530, Gangseo-gu in Busan (C3) with 3.312, Dong-gu in Daejeon (C2) with 3.164, and Gimhae-si in Gyeongnam (C3) with 3.115. Interestingly, while Gyeongsan-si in Gyeongbuk

| Classification | | Passenger travel | | | | Freight travel | | | |
|---|---|---|---|---|---|---|---|---|---|
| | | First central areas | | Second central areas | | Frist central areas | | Second central areas | |
| C1 | ① | Seoul Gangnamgu | 3.221 | Seoul Seochogu | 2.123 | Gyeonggi Bucheonsi | 4.354 | Gyeonggi Gimposi | 2.000 |
| | ② | Gyeonggi Seongnamsi Bundanggu | 3.093 | Gyeonggi Bucheonsi | 2.107 | Incheon Bupyeonggu | 3.530 | Incheon Seogu | 2.727 |
| | ③ | - | - | - | - | Seoul Gangseogu | 2.224 | Seoul Gurogu | 1.097 |
| C2 | ① | Jeonnam Najusi | 2.196 | Jeonnam Mokposi | 1.558 | Gyeongbuk Gyeongsansi | 3.716 | Gyeongbuk Yeongcheonsi | 1.558 |
| | ② | Gwangju Bukgu | 2.033 | Gwangju Gwangsangu | 1.396 | Daegu Donggu | 3.164 | Daegu Dalseogu | 1.964 |
| | ③ | - | - | - | - | Gwangju Gwangsangu | 2.041 | Gwangju Bukgu | 0.561 |
| | ④ | - | - | - | - | Ulsan Uljugun | 1.208 | Ulsan Namgu | 0.733 |
| | ⑤ | - | - | - | - | Jeonnam Gwangyangsi | 1.013 | Jeonnam Jangseonggun | 0.868 |
| | ⑥ | - | - | - | - | Jeonbuk Gunsansi | 0.304 | Jeonbuk Namwonsi | 0.293 |
| C3 | ① | Gyeongbuk Gyeongsansi | 4.610 | Gyeongbuk Chilgokgun | 1.169 | Busan Gangseogu | 3.312 | Busan Sasanggu | 0.819 |
| | ② | Daegu Suseonggu | 3.357 | Daegu Donggu | 2.420 | Gyeongnam Gimhaesi | 3.115 | Gyeongnam Yangsansi | 1.701 |
| | ③ | CyeongnamYangsansi | 0.676 | Gyeongnam Gimhaesi | 0.593 | - | - | - | - |
| | ④ | Busan Bukgu | 0.420 | Busan Geumjeonggu | 0.385 | - | - | - | - |
| C4 | ① | Chungbuk Jecheonsi | 1.455 | Chungbuk Cheongjusi Seowongu | 1.351 | Gangwon Wonjusi | 2.705 | Gangwon Yeongwolgun | 0.646 |
| | ② | Chungnam Cheonansi Seobukgu | 1.350 | Chungnam Cheonansi Dongnamgu | 1.098 | Chungbuk Chungjusi | 2.615 | Chungbuk Jecheonsi | 1.057 |
| | ③ | Gangwon Yeongwolgun | 1.273 | Gangwon Goseonggun | 1.052 | Chungnam Cheonansi Seobukgu | 0.226 | Chungnam Cheonansi Dongnamgu | 0.174 |
| C5 | ① | Sejong Sejongsi | 1.855 | - | - | Sejong Sejongsi | 1.789 | - | - |
| | ② | Daejeon Yuseonggu | 1.346 | Daejeon Seogu | 1.128 | Daejeon Yuseonggu | 1.016 | Daejeon Daedeokgu | 0.981 |
| | ③ | Jeonbuk Gunsansi | 0.022 | - | - | - | - | - | - |
| Comparison | |  | | | |  | | | |

**Fig 7. Comparison of power centrality in movements between si/do.**

has the most significant influence in passenger traffic, it ranks second in freight travel, following Gyeonggi. In Daegu, Suseong-gu serves as the central area for passenger traffic, while Dong-gu holds this position for freight travel. Similarly, in Gyeonggi, while Bundang-gu in Seongnam-si is the central area for passenger traffic, Bucheon-si takes this role for freight travel.

The centrality analysis of movements between si/do reveals a concentration of influence in regions C1, C2, and C3. This suggests that the primary and secondary central areas in these regions play a pivotal role in the transport and logistics network. Unexpectedly, power centrality was highly rated in Gyeongbuk's Gyeongsan-si and Jeonnam's Naju-si, which may be attributed to their unique geographical locations. Gyeongsan-si, due to its proximity to Daegu Metropolitan City, benefits from the economic and social advantages of the metropolitan area while maintaining its own industrial and cultural activities. Naju-si, home to the headquarters of the Korea Electric Power Corporation, has developed as a central hub for science, technology, and the energy industry, distinguishing it from other regions in Jeollanam-do. This suggests that high centrality in non-metropolitan regions may not only be a result of traditional transport infrastructure development or logistics system efficiency, but also be related to the concentrated accumulation of industrial and technological innovation, and the enhancement of intra- and inter-regional connectivity. Conversely, regions C4 and C5 exhibited relatively low centrality, indicating less developed connectivity and influence within the transport and logistics network. This could be due to a variety of factors, including inadequate transport infrastructure, inefficient logistics systems, and restrictions on economic activities, which may hinder regional development and population influx. To enhance the connectivity and influence of these regions, policy support is necessary for the expansion of transport infrastructure, the improvement of logistics systems, and the strengthening of interregional networks.

The second set of results, as shown in Fig 8, identifies the first and second central areas of passenger and freight travel clusters in various si/gun/gu regions. Gwangju and Ulsan were the regions where the first and second central areas for both passenger and freight travel were identical. In contrast, Chungnam had the same central areas, but the order was reversed. In cases where only one central area was the same for both types of travel, Daejeon had the same first central areas, while Gyeongbuk had the same first central area for passenger travel and the second for freight travel. Daegu had the same second central area for passenger travel and the first for freight travel. Gyeonggi and Chungbuk had the same second central areas for both types of travel. Despite these similarities, Seoul, Jeonnam, Gyeongnam, Busan, Gangwon, and Jeonbuk had completely different central areas for passenger and freight travel. Incheon had no comparison target as it only had clusters for freight travel.

Specifically, the differences in the influence and ranking of primary and secondary central areas in passenger and freight travel networks across various regions can be analyzed. In passenger travel, the primary central area was Daejeon Seo-gu (C5) with a score of 3.959, followed by Seoul Gangnam-gu (C1) with 3.527, Gwangju Buk-gu (C2) with 2.787, Busan Busanjin-gu (C3) with 2.611, Daegu Suseong-gu (C6) with 2.513, and Gangwon Sokcho-si (C4) with 2.271. In contrast, for freight travel, Gwangju Buk-gu (C2) had the highest score of 3.998, followed by Gyeongnam Gimhae-si (C3) with 3.701, Busan Gangseo-gu (C3) with 3.263, Chungnam Cheonan-si Dongnam-gu (C4) with 2.884, Incheon Namdong-gu (C1) with 2.584, and Gyeonggi Hwaseong-si (C1) with 2.268. Interestingly, Gwangju Buk-gu ranked third in passenger travel but first in freight travel.

In Busan, Busanjin-gu was the primary area for passenger travel, while Gangseo-gu was primary for freight travel. The centrality analysis of movements between regions indicates an even distribution within clusters for both passenger and freight travel. This suggests that the transport and logistics system functions effectively and extensively across the nation, without bias toward specific regions. Particularly, the high influence of Gwangju Buk-gu and Gwangsan-gu, Daejeon Seo-gu and Yuseong-gu, and Busan Busanjin-gu and Gangseo-gu, in both passenger and freight travel, underscores the centrality of transportation and goods movement in these regions. Therefore, it is crucial to enhance the network's efficiency and connectivity with surrounding areas via intensive monitoring of human and material flows.

| Classification | | Passenger travel | | | | Freight travel | | | |
|---|---|---|---|---|---|---|---|---|---|
| | | First central areas | | Second central areas | | Frist central areas | | Second central areas | |
| C1 | ① | Seoul Gangnamgu | 3.527 | Seoul Seochogu | 2.816 | Incheon Namdonggu | 2.584 | Incheon Junggu | 2.503 |
| | ② | Gyeonggi Seongnamsi Bungdanggu | 1.547 | Gyeonggi Bucheonsi | 0.852 | Gyeonggi Hwaseongsi | 2.268 | Gyeonggi Bucheonsi | 2.206 |
| | ③ | - | - | - | - | Seoul Gangseogu | 1.321 | Seoul Yangcheongu | 1.066 |
| C2 | ① | Gwangju Bukgu | 2.787 | Gwangju Gwangsangu | 2.785 | Gwangju Bukgu | 3.998 | Gwangji Gwangsangu | 3.915 |
| | ② | Jeonnam Najusi | 0.441 | Jeonnam Mokposi | 0.311 | Jeonnam Gwangyangsi | 1.221 | Jeonnam Jangseonggun | 0.849 |
| | ③ | - | - | - | - | Daejeon Seogu | 1.200 | Daejeon Daedeokgu | 1.065 |
| | ④ | - | - | - | - | Ulsan Namgu | 0.264 | Ulsan Junggu | 0.259 |
| | ⑤ | - | - | - | - | Jeonbuk Jeoneupsi | 0.209 | Jeonbuk Iksansi | 0.205 |
| C3 | ① | Busan Busanjingu | 2.611 | Busan Haeundaegu | 2.143 | Gyeongnam Gimhaesi | 3.701 | Gyeongnam Yangsansi | 2.625 |
| | ② | Ulsan Namgu | 1.824 | Ulsan Junggu | 1.317 | Busan Gangseogu | 3.263 | Busan Sasanggu | 1.596 |
| | ③ | Gyeongnam Changwonsi Seongsangu | 0.402 | Gyeongnam Changwonsi Uichanggu | 0.379 | Daegu Dalseogu | 2.237 | Daegu Dalseonggun | 1.763 |
| | ④ | - | - | - | - | Gyeongbuk Gumisi | 0.846 | Gyeongbuk Gyeongsansi | 0.823 |
| C4 | ① | Gangwon Sokchosi | 2.271 | Gangwon Goseonggun | 1.767 | Chungnam Cheonansi Dongnamgu | 2.884 | Chungnam Cheonansi Seobukgu | 2.806 |
| | ② | - | - | - | - | Chungbuk Cheongjusi Cheongwongu | 2.215 | Chungbuk Cheongjusi Seowongu | 2.177 |
| | ③ | - | - | - | - | Gangwon Wonjusi | 0.422 | Gangwon Gangneungsi | 0.311 |
| C5 | ① | Daejeon Seogu | 3.959 | Daejeon Yuseonggu | 3.263 | - | - | - | - |
| | ② | Chungnam Cheonansi Seobukgu | 0.943 | Chungnam Cheonansi Dongnamgu | 0.814 | - | - | - | - |
| | ③ | Jeonbuk Jeonjusi Deokjingu | 0.802 | Jeonbuk Jeonjusi Wansangu | 0.774 | - | - | - | - |
| | ④ | Chungbuk Cheongusi Heungdeokgu | 0.777 | Chungbuk Cheongjusi Seowongu | 0.736 | - | - | - | - |
| C6 | ① | Daegu Suseonggu | 2.513 | Daegu Dalseogu | 2.406 | - | - | - | - |
| | ② | Gyeongbuk Gyeongsansi | 1.071 | Gyeongbuk Chilgokgun | 0.469 | - | - | - | - |
| Comparison | |  The first central areas (the primary centers); The second central areas (the secondary centers) | | | |  The first central areas (the primary centers); The second central areas (the secondary centers) | | | |

**Fig 8. Comparison of power centrality in movements between si/gun/gu.**

In summary, metropolitan areas with high population density and well-developed transport infrastructure, such as Daejeon Seo-gu and Seoul Gangnam-gu, ranked high for passenger travel. These areas have a high demand for movement and serve as hubs of daily life and economic activities. For freight travel, regions with production and logistics facilities, such as Gwangju Buk-gu and Gyeongnam Gimhae-si, showed high influence. This indicates that freight travel is closely tied to the flow of economic activities, including production, processing, and distribution.

However, the primary and secondary central areas for passenger and freight travel were identical in some regions, such as Chungnam and Sejong, but differed in others, such as Seoul, Jeonnam, Gyeongnam, and Busan. Factors such as economic activities, industrial structures, and the respective regions' geographical characteristics may affect these variations.

## Discussion

This study analyzes inter-regional travel flows from an urban network perspective using passenger and freight travel data across South Korea, identifying key differences between passenger and freight travel patterns. The analysis of the origin and destination patterns of passenger and freight travel revealed that passenger transport occurs across a wider range of regions compared to freight travel, with a more pronounced imbalance between the volume of departures and arrivals.

Passenger travel primarily occurs for personal reasons, such as commuting, shopping, and leisure. Travel therefore occurs over a wide spatial range that includes not only densely populated large cities, but also medium-sized cities and surrounding areas. This perspective aligns with the findings presented in the study by Adams [3], which emphasizes that cities function as interconnected components of a larger system rather than existing independently. Understanding the relationships between cities is essential for effective urban planning and development. When the volume of arrivals exceeds that of departures in passenger travel, the region may be strengthening its role as a destination. This travel pattern also aligns with the view presented in Adams [3], which states that such interactions form a broad network. As demonstrated in Fig 9, the observed passenger travel patterns show dispersed and interconnected flows across regions, forming a cohesive and extensive travel network that supports Adams' assertion. This indicates that passenger travel largely relies on short-distance movements, forming dispersed travel routes for various purposes and thereby establishing an extensive travel network. In contrast, freight travel is concentrated around specific regions with active economic activities, such as industrial complexes, logistics centers, and ports, focusing on connections between specific routes and areas based on logistical efficiency and economic purposes. Freight travel reflects the physical logistics flow of moving goods from production sites to consumption areas, and this primarily takes place over long-distance routes. Regions that serve as hubs of economic activity tend to have higher departure volumes as they are actively involved in the connections between production, processing, and consumption. This aligns with Meijer's [27] urban network theory, which argues that inter-city cooperation optimizes logistics flows and functions as an efficient system. As shown in Fig 10, the freight

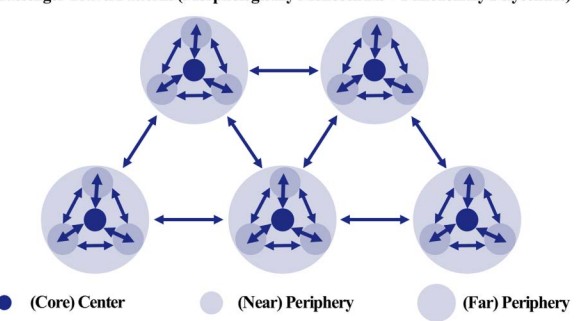

**Fig 9. Passenger travel patterns.**

**Freight Travel Patterns (Morphologically Polycentric + Functionally Monocentric)**

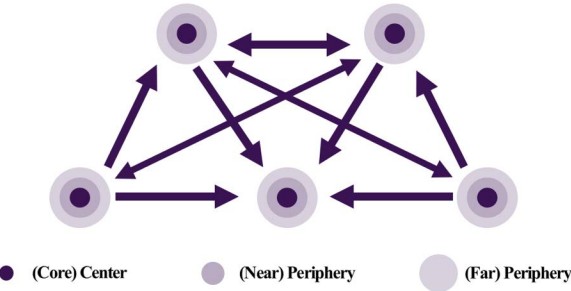

● (Core) Center ● (Near) Periphery ● (Far) Periphery

**Fig 10. Freight travel patterns.**

**Inter-Regional Passenger Travel Network Cluster Range**

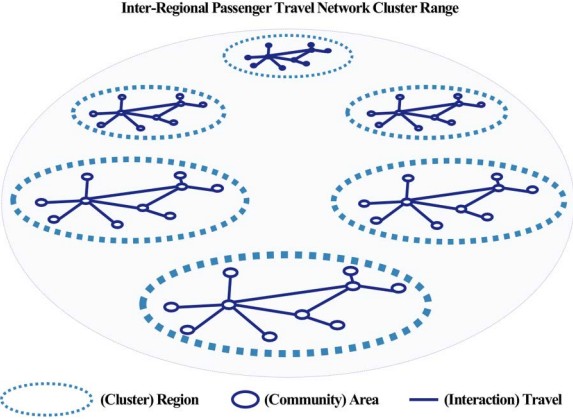

⬭ (Cluster) Region ◯ (Community) Area — (Interaction) Travel

**Fig 11. Passenger travel network clusters.**

travel patterns exhibit concentrated flows between major logistics hubs, demonstrating how inter-city collaboration enhances efficiency within the network.

It is also worth noting the reversal phenomenon observed between passenger and freight travel. In regions with active passenger travel, freight departure volumes tend to be relatively low, and vice versa. This phenomenon is closely linked to the economic characteristics and industrial structure of the respective regions. For example, densely populated areas primarily function as human resource centers, but if industries such as logistics and manufacturing are not well-developed, the volume of freight departures tends to be low. In contrast, in regions where agriculture or manufacturing is developed, freight travel tends to be more active, even with a relatively smaller population, as goods and resources produced in these areas are transported elsewhere. This aligns with the argument presented by Burger and Meijers [10], which suggests that inter-city networks are functionally complementary and can expand economic scale through cooperation. As highlighted in Fig 12, the complementary roles of passenger and freight travel clusters illustrate how distinct functional contributions enhance the overall economic integration of urban networks.

A comparison of the cluster formation ranges for passenger and freight travel using regular equivalence in network analysis revealed significantly different cluster formation patterns for the two types of travel. Freight travel clusters were formed over a wider geographic range compared to passenger travel clusters, but the number of clusters was

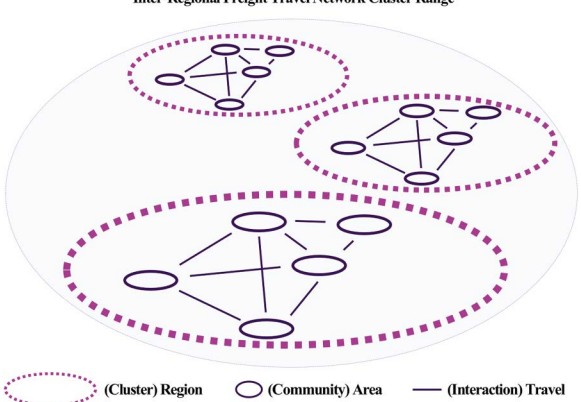

**Fig 12. Freight travel network clusters.**

relatively smaller. This is consistent with the findings of Frenken and Hoekman [4], who stated that urban networks focus on connecting specific routes and economic hubs. Our analysis of freight travel confirms this pattern, as major routes connecting industrial complexes, ports, and logistics centers serve as focal points within the network, as illustrated in Fig 12. This characteristic of freight travel is primarily centered around specific nodal areas, such as industrial complexes, ports, and logistics centers where logistics and economic activities are active. This aligns with Meijers' [27] explanation that urban network systems are structured based on cooperation between politically independent cities through transportation and communication infrastructure. In contrast, passenger travel tended to form more clusters within a smaller geographic range, possibly because passenger travel forms complex travel patterns that are driven by various individual purposes to different destinations, such as commuting, attending school, shopping, and leisure. These movements are largely influenced by individual decision-making and people's everyday activity ranges. This aligns with Batten [1], who stated that personal travel patterns in urban systems are dispersed, as well as Burger and Meijers [10], who suggested that passenger travel forms clusters due to connections to various destinations rather than concentrations along specific routes. As shown in Fig 11, passenger travel clusters are distributed over a relatively narrow range, reflecting the diverse purposes and localized nature of these trips. Therefore, while freight travel is concentrated on efficiently connecting specific regions that are closely linked to economic activities, passenger travel reflects more diverse and complex individual mobility patterns, resulting in a differentiated network structure with multiple clusters.

The power centrality analysis revealed a difference in the distance between the primary and secondary central areas for passenger travel and freight travel. The central areas of passenger travel are primarily formed around densely populated and economically active regions that have a relatively short distance between the primary and secondary central areas. This can be interpreted as passenger travel primarily relying on short-distance movements between regions, with relatively dense central areas forming due to the diverse travel purposes, such as commuting, shopping, and leisure. In contrast, freight travel tended to show a greater distance between the primary and secondary central areas. This is consistent with Capello's [28] assertion that freight travel follows long-distance economic flows, as well as González [18], who suggested that central areas are formed based on logistical efficiency and economic objectives. Fig 14 illustrates how the central areas of freight travel are situated in regions with developed

logistical infrastructure, reinforcing Capello's and González's arguments. The central areas for freight travel are located in regions with well-developed logistics and manufacturing activities, such as production sites, industrial complexes, logistics centers, and ports. These distances are shaped by logistical efficiency and economic objectives. Therefore, freight travel places importance on the connectivity between specific routes and economic hubs, which suggests that central areas can be formed over a wider range.

This difference is interpreted as stemming from the distinct purposes and requirements of passenger and freight travel. Passenger travel reflects short-distance movements within people's daily activity ranges, while freight travel aims to create economic value through the

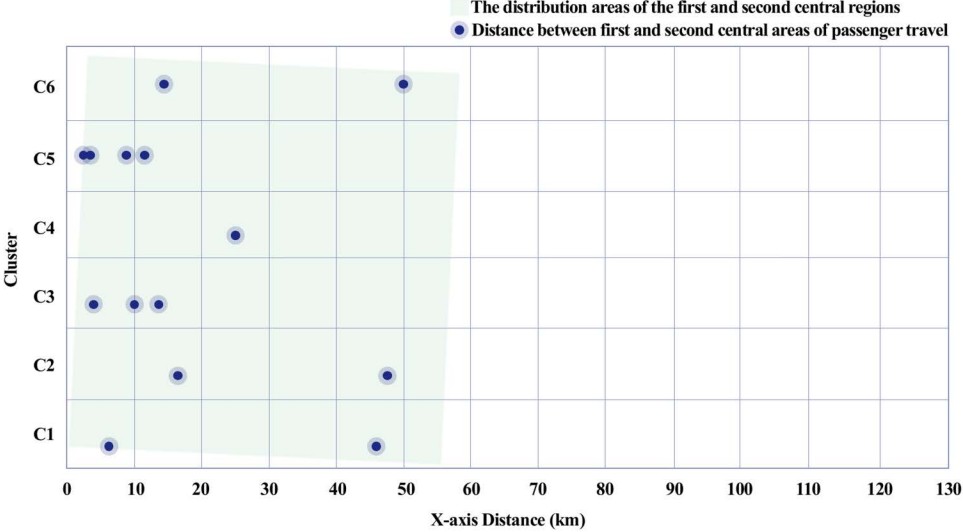

**Fig 13. Distance between central areas of the passenger travel network.**

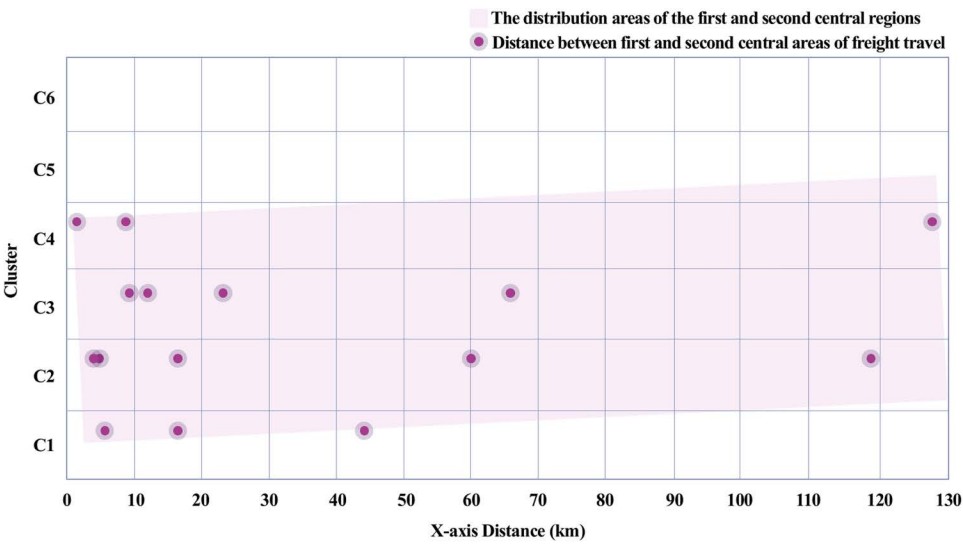

**Fig 14. Distance between central areas of the freight travel network.**

long-distance distribution of raw materials, manufactured goods, and consumer products. These differences in travel patterns demonstrate that the two modes of transport play distinct roles within the urban network. Fig 13 and Fig 14 show contrasting spatial distributions and centrality measures for passenger and freight travel, respectively, underscoring their distinct contributions to the network's economic efficiency, as also supported by prior studies [24,26,30].

## Conclusion

This study examines inter-regional travel flows in South Korea, focusing on passenger and freight travel patterns, central area formation, and strategies to improve regional connectivity. Through empirical data and network analysis, it identifies distinct characteristics of passenger and freight travel, highlighting their unique roles within the urban network.

First, passenger travel exhibited a higher volume and was characterized by short-distance movements that supported everyday activities such as commuting, shopping, and leisure. As shown in Fig 9, passenger travel formed dispersed and interconnected flows across regions, with densely populated areas often serving as central destinations due to a higher volume of arrivals than departures. In contrast, freight travel was concentrated in economically active regions, such as industrial complexes, logistics centers, and ports, with long-distance movements dominating the network. Fig 10 illustrates how freight travel primarily connects production sites to consumption areas, reflecting its role in optimizing logistics and supporting industrial activity.

Second, structural differences between the networks were evident. Passenger travel formed numerous clusters within smaller geographic ranges, as depicted in Fig 11, indicating diverse and localized mobility patterns. Freight travel, on the other hand, exhibited fewer clusters that spanned broader geographic areas, emphasizing efficient connections between economic hubs, as shown in Fig 12. These distinctions underscore that passenger travel supports localized social and economic interactions, while freight travel focuses on long-distance logistical efficiency.

Third, power centrality analysis revealed contrasting spatial distributions of central areas. Passenger travel central areas were densely distributed in regions with high population density and economic activity, with relatively short distances between primary and secondary centers (Fig 13). Conversely, freight travel demonstrated greater distances between central areas, highlighting its focus on connecting regions of production and consumption over long distances (Fig 14). These findings confirm that passenger travel enhances regional accessibility, while freight travel facilitates inter-regional economic integration.

These results underscore the necessity of developing complementary networks that integrate passenger and freight travel to achieve balanced regional development. Passenger travel networks, characterized by multi-nodal activity centers, should be strategically linked with freight travel hubs to optimize logistical flows and improve regional accessibility. For example, enhancing accessibility between densely populated areas and industrial complexes through integrated public transportation and logistics systems can address network inefficiencies and enhance regional productivity.

Additionally, the findings highlight the importance of planning for morphological and functional polycentricity. Passenger travel clusters, driven by diverse social activities, and freight travel clusters, centered on logistical efficiency, must be strategically coordinated to maximize regional productivity. Targeted investments in infrastructure and policies that strengthen these connections can foster balanced development between cities and regions.

In conclusion, this study provides empirical evidence of the distinct travel patterns and network roles of passenger and freight transport through detailed analysis and visualization. By aligning transportation and logistics strategies with these roles, policymakers can design

more integrated and sustainable regional planning frameworks. These findings serve as a critical foundation for evidence-based decision-making to enhance inter-regional interactions and promote balanced development across South Korea.

## Limitations and future research

This study provides foundational data for enhancing regional transport infrastructure and economic activities by distinguishing regions where the central areas of passenger and freight travel do and do not coincide. This distinction allows for an examination of connectivity between densely populated and economically active areas by identifying differing movement trends in passenger and freight travel. A quantitative assessment of these central areas and their influence can inform policy decisions aimed at constructing an efficient urban network and optimizing logistics systems. This study also offers insights into the complex dynamics of passenger and freight travel, which are integral to various social and economic activities, and proposes an analysis for developing an efficient transport and logistics system from an urban network perspective. However, this study, which analyzes interregional movement patterns and central areas of passenger and freight travel, has certain limitations. First, it relies on single-year data from 2019, which, while offering a snapshot in time, does not capture the dynamic trends of changing travel patterns over time.

Second, the analysis method, which uses total travel volume data, provides broad-scale results but lacks depth in understanding detailed aspects such as specific purposes or travel volume in specific regions. Finally, the spatial scope of the study is limited. While the analysis at the level of si/gun/gu reveals broad regional characteristics, it fails to identify microscopic network characteristics at a more detailed level.

To address these limitations, future research should consider the following directions: examining changes in socioeconomic situations and the resulting flows in human and material networks using multi-year data to understand more dynamic travel patterns; conducting more granular analyses, such as passenger travel according to purpose, freight travel based on resource movement, and micro-geographic units; and undertaking additional research on transport connectivity and economic revitalization in areas with relatively low centrality to promote balanced regional development. This will establish a basis for policy recommendations to address regional imbalance and propose efficient strategies for developing shrinking cities.

## Acknowledgments

The authors thank the anonymous reviewers and editors for their valuable and constructive suggestions for improving this article.

## Author contributions

**Conceptualization:** Soyeong Lee.

**Data curation:** Soyeong Lee.

**Formal analysis:** Soyeong Lee.

**Funding acquisition:** Heesun Joo.

**Investigation:** Soyeong Lee.

**Methodology:** Soyeong Lee.

**Project administration:** Heesun Joo.

**Resources:** Heesun Joo.

**Supervision:** Heesun Joo.

**Validation:** Soyeong Lee.

**Visualization:** Soyeong Lee.

**Writing – original draft:** Soyeong Lee.

**Writing – review & editing:** Heesun Joo.

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
