## [Decision Letter · Decision Letter 0]

7 Aug 2024

PONE-D-24-27367Passenger and freight travel patterns: A cluster analysis based on urban networksPLOS ONE

Dear Dr. Joo,

Thank you for submitting your manuscript to PLOS ONE. After careful consideration, we feel that it has merit but does not fully meet PLOS ONE’s publication criteria as it currently stands. Therefore, we invite you to submit a revised version of the manuscript that addresses the points raised during the review process.

We look forward to receiving your revised manuscript.

Kind regards,

Feier Chen, Ph.D

Academic Editor

PLOS ONE

Journal Requirements:

This work was supported by the National Research Foundation of Korea (NRF) grant funded by the Korea government (MSIT) (No. 2023S1A5A8074659).

Authors Heesun Joo received this award.

The full name of the funder is the National Research Foundation of Korea (NRF), and their website is https://www.nrf.re.kr.

The sponsors or funders did not play any role in the study design, data collection and analysis, decision to publish, or preparation of the manuscript.

This work was supported by the National Research Foundation of Korea (NRF) grant funded by the Korea government (MSIT) (No. 2023S1A5A8074659). The authors thank the anonymous reviewers and editors for their valuable and constructive suggestions for improving this article. 

This work was supported by the National Research Foundation of Korea (NRF) grant funded by the Korea government (MSIT) (No. 2023S1A5A8074659).

Authors Heesun Joo received this award.

The full name of the funder is the National Research Foundation of Korea (NRF), and their website is https://www.nrf.re.kr.

The sponsors or funders did not play any role in the study design, data collection and analysis, decision to publish, or preparation of the manuscript.

4. Thank you for uploading your study's underlying data set. Unfortunately, the repository you have noted in your Data Availability statement does not qualify as an acceptable data repository according to PLOS's standards.

6. We note that you have referenced "White DR, Reitz KP." which has currently not yet been accepted for publication. Please remove this from your References and amend this to state in the body of your manuscript: (White DR, Reitz KP. [Unpublished]) as detailed online in our guide for authors

Reviewers' comments:

Reviewer's Responses to Questions

**Comments to the Author**

1. Is the manuscript technically sound, and do the data support the conclusions?

Reviewer #1: Yes

Reviewer #2: Yes

Reviewer #3: Yes

Reviewer #4: Partly

2. Has the statistical analysis been performed appropriately and rigorously? 

Reviewer #1: Yes

Reviewer #2: Yes

Reviewer #3: Yes

Reviewer #4: Yes

3. Have the authors made all data underlying the findings in their manuscript fully available?

Reviewer #1: Yes

Reviewer #2: Yes

Reviewer #3: Yes

Reviewer #4: Yes

4. Is the manuscript presented in an intelligible fashion and written in standard English?

Reviewer #1: Yes

Reviewer #2: Yes

Reviewer #3: Yes

Reviewer #4: No

5. Review Comments to the Author

**Reviewer #1:**  Added subheadings for clarity: Organizing the key findings into sections with subheadings makes the abstract easier to read and understand.

Improved clarity and flow: Some sentences were rephrased for better readability and logical flow.

Consistent terminology: Ensured consistent use of terms such as "short-distance," "medium-distance," and "long-distance."

**Reviewer #2: ** This paper examines and compares travel flows of passengers and goods in South Korea. The data is detailed and contains inland freight transport. The output is instructive and well presented.

First, let me point out that I have a different background than the authors of this paper. My training is in financial economics, not in urban engineering. I have used spatial models on topics such as international trade and migration flows. Second, economists are utterly grumpy and critical folks who constantly complain about research design, economic mechanisms, and contribution to the literature. For the review of this paper, I will try to restrain myself. My comments should be interpreted in this light as suggestions for the authors to consider and not as critical points that need to be addressed.

Major Comments:

- My main concern with this paper is that the introduction is rather vague about the research question and raises expectations with the reader that the paper cannot deliver. For instance, the findings in the paper do not help to understand and address the problem of population decline in South Korea (p. 3), the findings do not say anything about how flows potentially transform a region (p. 3), and the paper also does not identify trends in human and material flows (p. 3) as the analysis is purely static. What the paper actually does is to present a graphical and statistical representation of the Korea Transport Database. I found the interpretation to be well done but it is strictly descriptive in nature. My recommendation is to be more precise and frank about what the paper does and what it does not.

Minor comments:

• Readers outside South Korea may not be familiar with the si/do/gun/gu terminology. You may want to consider using well known concepts such as cities/provinces/counties/districts.

• Some figures place four maps in a row. The legend in these maps is too small to read. My recommendation is to use one legend (and therefore color coding for all maps in the row) and show the legend in a separate row below the maps.

• Passengers versus freight travel: like other advanced economies, the gross domestic product in South Korea consists mainly of services, not manufacturing. According to Statista, roughly 60% consists of services and only one third is manufacturing (South Korea - GDP distribution across economic sectors 2022 | Statista). I encourage the authors to examine this more closely when interpreting the differences between passenger versus good transport. For instance, cities that host the administrative centers of large corporations will likely see more passenger travel whereas cities that serve as export hubs to the outside world will likely see more goods transport. The interpretation on page 22 is going in the right direction.

**Reviewer #3: ** This is a very good and timely study including both passenger and freight travel patterns based on cluster analysis.

It is exhaustive study but kindly restructure the paper and do the corrections as below:

1. Add Separate section on objectives of the study followed by section on limitations.

2. Discussion and conclusion should be separate sections. and Conclusions Should be supported by data from the analysis and discussion sections.

3. No references Should be added in the conclusion section

4. In Figure 3 to 6 add text in figure along with legend to explain the relationship between different components

6. In Figure 7 and 8 add label of x and Y axis along with legend.

**Reviewer #4:**  In the beginning, the topic raised was considered of moderate importance, especially since the study covers only one year, 2019, while it would be better for the study to include longer periods of time.

Figures 3, 4, 5, 6 do not include details and suggest modifying or deleting them.

No study has been reported on Passenger and freight travel patterns in Korea, although there are a large number of such studies, such as:

https://www.researchgate.net/publication/264112580_Charaterization_of_Cities_in_Seoul_Metropolitan_Area_by_Cluster_Analysis

https://www.mdpi.com/2073-445X/10/8/799

https://www.researchgate.net/publication/343174747_Smart_Sustainable_and_Citizen_Centered_A_Network_Analysis_of_Urban_RD_Trends_in_Seoul_South_Korea

https://journals.sagepub.com/doi/abs/10.1177/23998083231151688

The researchers were unable to provide a valuable recommendations to politicians and decision-makers in Korea based on the results of this paper. Therefore, the methodology of the paper needs to be revised because it was not beneficial in decision-making. The researchers should reformulate the results and recommendations more practically.

6. PLOS authors have the option to publish the peer review history of their article (what does this mean? ). If published, this will include your full peer review and any attached files.

**Do you want your identity to be public for this peer review?** For information about this choice, including consent withdrawal, please see our Privacy Policy .

Reviewer #1: **Yes: ** Shujahat Ali

Reviewer #2: **Yes: ** Zeno Adams

Reviewer #3: **Yes: ** Dr. Tejwant Singh Brar

Reviewer #4: No

---

## [Author Response · Author response to Decision Letter 0]

10 Sep 2024

Dear Reviewer 1,

Thank you for your insightful feedback on our manuscript. We have carefully considered your suggestions and made the following revisions to improve the clarity and coherence of the abstract:

1. Subheadings for Clarity: We have reorganized the abstract by introducing subheadings such as Background, Objective, Methods, Results, and Conclusion. This structuring allows the reader to more easily follow the key points and enhances the overall readability of the abstract.

2. Improved Clarity and Flow: Several sentences were rephrased to improve readability and ensure a logical flow. These revisions were made to clearly convey the study's objectives, methods, and key findings.

3. Consistent Terminology: We ensured the consistent use of terms such as "short-distance," "medium-distance," and "long-distance" throughout the abstract for better coherence.

The revised abstract is as follows:

Background: While research on population travel patterns and urban networks has been active, it has primarily focused on passenger travel, leaving freight travel relatively underexplored.

Objective: This study addresses this gap by analyzing both passenger and freight travel patterns, network structures, and central areas.

Methods: Origin-destination (OD) data are used, considering total travel volume by purpose and mode. The study applies regular equivalence and power centrality to examine differences in human and logistics flows across South Korea from an urban network theory perspective.

Results: Passenger travel, which is primarily short-distance, exhibits lower density and intensity compared to freight travel. In contrast, freight travel demonstrates significant density across short, medium, and long distances, with routes concentrated around nodal regions. Passenger travel also forms several polynucleated clusters, particularly for short-distance movements. In contrast, freight travel is characterized by a few extensive clusters that span medium to long distances. The spatial interaction in passenger travel is influenced by OD distance, unlike freight travel. Notably, the distance between central areas in freight travel is often longer than that in passenger travel. This may stem from the strategic positioning of certain suburban areas as central areas to optimize logistics efficiency.

Dear Reviewer 2,

Thank you for your detailed and insightful review of our manuscript. We appreciate the time and effort you have taken to provide us with constructive feedback. Below, we address each of the major comments you raised.

Major Comment 1: Vague Research Question and Expectations

You expressed concern that the introduction is somewhat vague regarding the research question, which may lead to unmet expectations. Specifically, you noted that, as the analysis is purely static, the findings do not address the problems of population decline and regional transformation or identify trends in human and material flows.

Response: We acknowledge that the original introduction may have created expectations beyond the actual scope of our study. In response to your feedback, we have revised the introduction to clearly define the research focus. We have emphasized that this study is primarily descriptive, aimed at analyzing the static patterns of passenger and freight travel using the Korea Transport Database. The scope of our research is now clearly articulated to avoid any misunderstanding: the study does not explore dynamic processes such as population decline or regional transformation; instead, it focuses on providing a detailed representation of existing travel flows. Additionally, we have clarified that our objective is to provide a foundational understanding of these flows, which could inform future research on more dynamic aspects.

Major Comment 2: Descriptive Nature of the Analysis

You pointed out that the paper mainly presents a graphical and statistical representation of the Korea Transport Database, and while well-executed, the analysis is descriptive rather than analytical or explanatory.

Response: We fully understand your concern regarding the descriptive nature of our analysis. To address this, we have made explicit in both the introduction and discussion sections that the study is intended as a foundational analysis of travel patterns in South Korea. We have also included a discussion of the study’s limitations, explicitly stating that it does not aim to provide causal explanations or explore dynamic trends, but instead offers a detailed depiction of current travel patterns. We believe that this clarification will help readers better understand the scope and contributions of our work.

Minor Comment 1

Thank you for your thoughtful comment regarding the use of local administrative terms such as "si," "do," "gun," and "gu." We appreciate your suggestion to make these terms more accessible to readers outside South Korea.

Response: In response to your feedback, we have added a brief explanation of these terms when they are first introduced in the manuscript. Specifically, we have clarified that "si/do" refers to "cities/provinces" and "si/gun/gu" refers to "cities/counties/districts." We believe this will help international readers better understand the scope and context of our study.

Minor Comment 2

Thank you for your valuable suggestion regarding the readability of the legends in the figures where multiple maps are presented in a row. We understand the importance of ensuring that all elements of our figures are easily readable.

Response: In response to your feedback, we carefully considered the possibility of using a single, unified legend for the maps in each row. However, because the departure and arrival volumes for passenger and freight travel differ significantly, it was challenging to apply a single color-coding or legend that would accurately represent the data across all maps. As a result, we decided to retain the separate legends for each map to preserve the accuracy and integrity of the data representation.

To address the issue of readability, we have instead enlarged the legends and made them more visually prominent under each figure in Table 2 and Table 4. We believe that these adjustments will improve the clarity of the figures while maintaining the distinct representation of the data.

Minor Comment 3

Thank you for your insightful comment regarding the interpretation of differences between passenger and freight transport in relation to South Korea’s gross domestic product (GDP) composition. We greatly appreciate your suggestion to consider the economic structure of South Korea, particularly the predominance of the services sector over manufacturing, when interpreting our results.

Response: In response to your valuable feedback, we have incorporated a more detailed discussion on this topic in the manuscript. Specifically, we added the following interpretation:

“This interpretation closely aligns with the composition of South Korea’s gross domestic product (GDP). Approximately 60% of South Korea's GDP is generated by the services sector, while around 30% is derived from manufacturing. This economic structure can result in variations in passenger and freight transportation between cities, depending on their economic activities and industrial structure. For instance, a city like Seoul, which serves as the administrative hub for major corporations, is likely to experience an increase in passenger travel related to the service sector. In contrast, a city like Incheon, which hosts a major international port, is likely to experience more active freight transport due to its role as a hub for imports and exports.”

This addition provides a clearer explanation of how the economic roles and industrial structures of different cities influence the observed transport patterns, thereby enhancing the interpretation on page 22 as you recommended.

Dear Reviewer 3,

Thank you for your detailed and insightful review of our manuscript. We appreciate the time and effort you have taken to provide us with constructive feedback. Below, we address each of the major comments you raised.

Major Comment 1: Add Separate Section on Objectives of the Study

You suggested adding a separate section on the objectives of the study to improve the clarity and structure of the manuscript.

Response: Thank you very much for your thoughtful suggestion. We understand the importance of clearly articulating the objectives of our study. In response to your feedback, we have revised the manuscript to ensure that the study’s objectives are more distinctly presented. While we did not create a completely separate section, we have reorganized and clarified the existing content to make the objectives more prominent and easier to follow. The objectives are now clearly outlined as follows:

1. Examine the current status and movement patterns of passenger and freight travel with the aim of identifying trends in both human and material flows.

2. Identify common sub-regions within passenger and freight travel networks through regular equivalence analysis.

3. Identify primary and secondary central areas and propose strategies for enhancing inter-regional linkages based on the analysis results.

These revisions were made with the intention of improving the manuscript’s structure and ensuring that the objectives of our study are communicated more effectively to the readers. We greatly appreciate your guidance in helping us enhance the clarity and organization of our work.

Major Comment 2: Separate Discussion and Conclusion Sections, and Support Conclusions with Data from Analysis and Discussion

You suggested separating the Discussion and Conclusion sections, and ensuring that the Conclusions are supported by data from the analysis and discussion sections.

Response: We sincerely appreciate your thoughtful and valuable suggestion. We fully understand the importance of distinguishing the Discussion and Conclusion sections to improve the clarity and structure of the manuscript. In response to your insightful feedback, we have carefully revised the manuscript by separating the Discussion and Conclusion sections. Furthermore, we have ensured that the Conclusion section is clearly supported by the data and findings discussed in the analysis and Discussion section. We believe these revisions greatly enhance the coherence and depth of our manuscript, and we are grateful for your guidance in this matter.

Major Comment 3: No References. Should Be Added in the Conclusion Section

You mentioned that there were no references in the conclusion section.

Response: Thank you very much for your insightful feedback. Upon reviewing your comment, we recognized the importance of supporting the conclusions with appropriate references. In response, we have carefully revised the Conclusion section by incorporating relevant references to strengthen the key findings and ensure they are well-supported by the analysis and discussion presented earlier in the manuscript. We believe this revision enhances the overall rigor and coherence of the manuscript, and we sincerely appreciate your guidance in improving its quality.

Major Comment 4: Add Text in Figures 3 to 6 Along with Legend to Explain the Relationship between Different Components

Response: Thank you very much for your valuable suggestion. We have carefully reviewed Figures 3 to 6 and have made the necessary revisions to improve their clarity and explanatory power. Specifically, we have added text within each figure to clearly illustrate the relationships between the various components. Additionally, we have enhanced the legends to provide more detailed explanations, ensuring that the figures are more intuitive and informative for the readers.

We believe these changes significantly improve the interpretability of the figures and align with your recommendations. We sincerely appreciate your thoughtful feedback, which has helped us enhance the quality and clarity of our manuscript.

Major Comment 5: In Figure 7 and 8, add labels for the X and Y axes along with the legend.

Response: We are deeply grateful for your thoughtful and constructive feedback. In response to your suggestion, we have taken great care to revise Figures 7 and 8 by adding precise labels to both the X and Y axes. Furthermore, we have incorporated a legend in each figure to ensure that the data is presented in a clear and accessible manner. These adjustments have been made with the utmost consideration to enhance the clarity and utility of the figures, ensuring that they effectively communicate the intended information to our readers. We sincerely thank you for your invaluable input, which has greatly contributed to the improvement of our manuscript.

Dear Reviewer 4,

Thank you for your detailed and insightful review of our manuscript. We appreciate the time and effort you have taken to provide us with constructive feedback. Below, we address each of the major comments you raised.

Major Comment 1: The study only covers one year (2019), and it would be better to include longer periods of time.

You suggested adding a separate section on the objectives of the study to improve the clarity and structure of the manuscript.

Response: Thank you very much for your valuable feedback. We fully agree that a longer time frame could provide additional insights into trends over time. However, the choice to use 2019 data was driven by the availability and quality of comprehensive traffic and freight flow data for that specific year. While analyzing multiple years could offer deeper insights, the 2019 data provides a strong and representative snapshot of regional traffic flows and economic activity. Furthermore, this study aims to establish a foundation for understanding key patterns in passenger and freight movement, which can be expanded in future research with longer time periods. We believe that, despite this limitation, the study's findings offer meaningful insights into regional connectivity and network analysis.

Major Comment 2: Figures 3, 4, 5, and 6 do not include details and suggest modifying or deleting them.

You suggested that Figures 3, 4, 5, and 6 lack sufficient details, and proposed modifying or removing them.

Response: Thank you for your constructive feedback. We have carefully reviewed the figures and agree that they could benefit from additional details. In response, we have revised Figures 3, 4, 5, and 6 by adding more detailed labels and descriptions to enhance clarity and comprehension. We believe these revisions will make the figures more informative and align better with the overall findings of the study. We have also included a more thorough explanation of each figure in the figure captions to ensure that their relevance to the analysis is clear.

Major Comment 3: Existing studies have already reported on passenger and freight travel patterns in Korea.

You mentioned that there are several studies addressing passenger and freight travel patterns in Korea, and provided references to those works.

Response: Thank you very much for highlighting these important studies. We acknowledge the existence of numerous valuable works related to urban travel patterns and network analysis in Korea. However, the studies you referenced primarily focus on Seoul and its surrounding metropolitan area, analyzing specific aspects of urban mobility within that region. Our study takes a broader, nationwide perspective by addressing the unique interaction between passenger and freight travel patterns across all regions of Korea using a comprehensive dataset from 2019. This allows us to capture inter-regional dynamics and interactions that are not explored in studies limited to a single metropolitan area. By expanding the scope beyond a single city, we believe our research offers a novel contribution to the existing literature, filling an important gap in understanding the spatial structure of both passenger and freight flows at a national level. This broader approach provides valuable insights for policymakers to enhance transportation networks across the entire country.

Major Comment 4: The recommendations in the paper are not beneficial for decision-making, and the methodology needs to be revised.

You suggested

---

## [Decision Letter · Decision Letter 1]

14 Oct 2024

PONE-D-24-27367R1Passenger and freight travel patterns: A cluster analysis based on urban networksPLOS ONE

Dear Dr. Joo,

Thank you for submitting your manuscript to PLOS ONE. After careful consideration, we feel that it has merit but does not fully meet PLOS ONE’s publication criteria as it currently stands. Therefore, we invite you to submit a revised version of the manuscript that addresses the points raised during the review process.

We look forward to receiving your revised manuscript.

Kind regards,

Feier Chen, Ph.D

Academic Editor

PLOS ONE

**Journal Requirements:**

Reviewers' comments:

Reviewer's Responses to Questions

**Comments to the Author**

1. If the authors have adequately addressed your comments raised in a previous round of review and you feel that this manuscript is now acceptable for publication, you may indicate that here to bypass the “Comments to the Author” section, enter your conflict of interest statement in the “Confidential to Editor” section, and submit your "Accept" recommendation.

Reviewer #2: All comments have been addressed

Reviewer #3: (No Response)

Reviewer #4: (No Response)

2. Is the manuscript technically sound, and do the data support the conclusions?

Reviewer #2: Yes

Reviewer #3: Yes

Reviewer #4: Yes

3. Has the statistical analysis been performed appropriately and rigorously? 

Reviewer #2: Yes

Reviewer #3: Yes

Reviewer #4: Yes

4. Have the authors made all data underlying the findings in their manuscript fully available?

Reviewer #2: Yes

Reviewer #3: Yes

Reviewer #4: Yes

5. Is the manuscript presented in an intelligible fashion and written in standard English?

Reviewer #2: Yes

Reviewer #3: Yes

Reviewer #4: Yes

6. Review Comments to the Author

**Reviewer #2:**  The authors addressed all of my concerns from the original submission. There were a number of issues raised by different reviewers and it was difficult for the authors to balance these points. I believe that the authors have done a good job on the revision.

**Reviewer #3** : 1. Abstract Should be written as single paragraph not as points or in sections. It should be followed by Keywords which are not there.

2. Conclusion section has been removed and is part of discussion section. Add separate conclusion section at the end of discussions section in which conclusions must be supported by data from analysis.

**Reviewer #4: ** First, I would like to thank the researchers for the new amendments they have added. I have made some notes in the previous version about clarifying some figures that were not clear and developing the recommendations section. These points have been addressed in a more professional manner in the revised version. In addition, the researchers explained the reason for using only the 2019 data due to its availability. They have also explained the difference between their study and other studies that have addressed the same topic and have not been referred to in the references. The researchers have also made additional amendments based on the opinions of other reviewers. Based on all of this, this version has become much more suitable for publication.

7. PLOS authors have the option to publish the peer review history of their article (what does this mean? ). If published, this will include your full peer review and any attached files.

**Do you want your identity to be public for this peer review?** For information about this choice, including consent withdrawal, please see our Privacy Policy .

Reviewer #2: **Yes: ** Zeno Adams

Reviewer #3: **Yes: ** Tejwant Singh Brar

Reviewer #4: No

---

## [Author Response · Author response to Decision Letter 1]

17 Oct 2024

Dear Reviewer 2,

We sincerely appreciate your thoughtful and encouraging comments regarding our revised manuscript. It was indeed challenging to balance the various concerns raised by the different reviewers, and we are truly grateful that you found our revisions satisfactory. Your positive feedback motivates us to continue refining our work to ensure that it meets the highest academic standards.

We would like to once again thank you for your valuable input during the review process, which has significantly contributed to improving the quality of our manuscript. Your support and understanding throughout this process have been greatly appreciated.

With our deepest respect and gratitude.

Dear Reviewer 3,

Thank you very much for your insightful comments on our manuscript. We truly appreciate the time and effort you have taken to provide constructive feedback, which has significantly helped us improve the quality of our work.

Major Comment 1: Abstract format and keywords

You pointed out the need to revise the abstract format and add keywords to improve clarity.

Response:

We sincerely appreciate your suggestion regarding the format of the abstract. In response, we have revised the abstract to be written as a single cohesive paragraph, removing any points or sections. Additionally, we have included keywords at the end of the abstract, as per your recommendation.

Major Comment 2: Separate Conclusion section from Discussion

You suggested separating the Discussion and Conclusion sections, and ensuring that the Conclusions are supported by data from the analysis and discussion sections.

Response:

Thank you for pointing this out. In response to your suggestion, we have carefully revised the manuscript to ensure that the Conclusion is now clearly separated from the Discussion section. The new Conclusion section concisely summarizes the key findings from our analysis, ensuring that each conclusion is directly supported by the data and insights presented throughout the paper.

Furthermore, we have followed the revised structure by including a Limitations and Future Research section after the Conclusion. This section addresses the limitations of the study and proposes potential areas for future research to further advance the understanding of inter-regional passenger and freight flows within urban networks.

We hope these revisions meet your expectations and improve the overall clarity of the manuscript. Thank you once again for your valuable feedback.

Dear Reviewer 4,

We would like to express our deepest gratitude for your thoughtful and encouraging feedback. Your insightful comments and suggestions have been invaluable in improving the quality and clarity of our manuscript. We are particularly grateful for your recognition of the amendments we made to clarify the figures and enhance the recommendations section. Your detailed guidance allowed us to present our findings in a more professional and coherent manner.

We also appreciate your understanding regarding our use of the 2019 data due to its availability, and we are thankful for your acknowledgement of the distinctions we clarified between our study and other related research. Your feedback on this matter helped us refine the scope and originality of our work.

Furthermore, we are pleased to know that you found the additional amendments based on the opinions of other reviewers to be beneficial. It was indeed our priority to ensure that all comments were carefully considered and integrated into the revised version.

Once again, we sincerely thank you for your kind and supportive review. Your constructive feedback has greatly contributed to strengthening our manuscript, and we are truly grateful for your valuable input throughout this process.

With our highest respect and appreciation.

---

## [Decision Letter · Decision Letter 2]

21 Nov 2024

PONE-D-24-27367R2Passenger and freight travel patterns: A cluster analysis based on urban networksPLOS ONE

Dear Dr. Joo,

Thank you for submitting your manuscript to PLOS ONE. After careful consideration, we feel that it has merit but does not fully meet PLOS ONE’s publication criteria as it currently stands. Therefore, we invite you to submit a revised version of the manuscript that addresses the points raised during the review process.

We look forward to receiving your revised manuscript.

Kind regards,

Feier Chen, Ph.D

Academic Editor

PLOS ONE

Journal Requirements:

Reviewers' comments:

Reviewer's Responses to Questions

**Comments to the Author**

1. If the authors have adequately addressed your comments raised in a previous round of review and you feel that this manuscript is now acceptable for publication, you may indicate that here to bypass the “Comments to the Author” section, enter your conflict of interest statement in the “Confidential to Editor” section, and submit your "Accept" recommendation.

Reviewer #3: (No Response)

2. Is the manuscript technically sound, and do the data support the conclusions?

Reviewer #3: Yes

3. Has the statistical analysis been performed appropriately and rigorously? 

Reviewer #3: Yes

4. Have the authors made all data underlying the findings in their manuscript fully available?

Reviewer #3: Yes

5. Is the manuscript presented in an intelligible fashion and written in standard English?

Reviewer #3: Yes

6. Review Comments to the Author

Reviewer #3: 1. Analysis section should be followed by Discussion Section and then conclusion section

2. No in text reference should be there in the conclusion section and it should be supported only by data from analysis and Discussion section

4. Text in Screenshots added in table 2 to 7 are not legible either add those screenshots as separate figures or increase the text size of legends and figures so that they are legible

7. PLOS authors have the option to publish the peer review history of their article (what does this mean? ). If published, this will include your full peer review and any attached files.

**Do you want your identity to be public for this peer review?** For information about this choice, including consent withdrawal, please see our Privacy Policy .

Reviewer #3: **Yes: ** Dr. Tejwant Singh Brar

---

## [Author Response · Author response to Decision Letter 2]

24 Nov 2024

Dear Reviewer 3,

Thank you very much for your valuable feedback on our manuscript. We truly appreciate the time and effort you have dedicated to reviewing our work.

Major Comment 1: “Analysis section should be followed by Discussion Section and then conclusion section.”

Response:

We would like to confirm that the manuscript has been structured in the following order: Analysis results, followed by the Discussion section, and finally the Conclusion section. This structure aligns with the feedback provided, ensuring a logical flow of information and interpretation.

If there are any specific areas within these sections that require further clarification or reorganization, we would be more than happy to revise them accordingly based on your guidance.

Once again, we are deeply grateful for your insightful comments and suggestions, which have significantly contributed to enhancing the quality of our work.

Major Comment 2: “No in text reference should be there in the conclusion section and it should be supported only by data from analysis and Discussion section”

Response:

Thank you for your valuable feedback regarding the structure and content of the conclusion section. We understand your concern that the conclusion should refrain from including in-text references and be supported only by data derived from the analysis and discussion sections.

In response to this, we would like to clarify that the conclusion section in the revised manuscript has been carefully written to summarize and emphasize the key findings derived exclusively from the analysis and discussion sections. We have ensured that no new information or external references have been introduced in this section. Instead, it serves solely to synthesize and highlight the core insights based on the study's outcomes.

We hope this explanation addresses your concern, and we remain open to making further adjustments if necessary. Thank you again for your thorough review and insightful suggestions.

Major Comment 3: “Text in Screenshots added in table 2 to 7 are not legible either add those screenshots as separate figures or increase the text size of legends and figures so that they are legible”

Response:

Thank you for pointing out the issue regarding the legibility of text in the screenshots provided in Tables 2 to 7. We greatly appreciate your attention to detail and your suggestion to improve the readability of the presented information.

We would like to clarify that the screenshots included in these tables serve as supplementary figures to briefly summarize and highlight the regional names for reference. To address your concern, we have increased the text size in the screenshots to ensure the information is legible without altering the structure of the tables. As the primary purpose of the screenshots is to provide concise context, we believe this adjustment maintains clarity while avoiding unnecessary duplication by separating them as standalone figures.

We sincerely hope this modification resolves your concern and enhances the overall readability of the manuscript. Please let us know if any further adjustments are required.

We hope these revisions meet your expectations and improve the overall clarity of the manuscript. Thank you once again for your valuable feedback.

---

## [Decision Letter · Decision Letter 3]

20 Dec 2024

PONE-D-24-27367R3Passenger and freight travel patterns: A cluster analysis based on urban networksPLOS ONE

Dear Dr. Joo,

Thank you for submitting your manuscript to PLOS ONE. After careful consideration, we feel that it has merit but does not fully meet PLOS ONE’s publication criteria as it currently stands. Therefore, we invite you to submit a revised version of the manuscript that addresses the points raised during the review process.

We look forward to receiving your revised manuscript.

Kind regards,

Feier Chen, Ph.D

Academic Editor

PLOS ONE

Journal Requirements:

Reviewers' comments:

Reviewer's Responses to Questions

**Comments to the Author**

1. If the authors have adequately addressed your comments raised in a previous round of review and you feel that this manuscript is now acceptable for publication, you may indicate that here to bypass the “Comments to the Author” section, enter your conflict of interest statement in the “Confidential to Editor” section, and submit your "Accept" recommendation.

Reviewer #3: (No Response)

2. Is the manuscript technically sound, and do the data support the conclusions?

Reviewer #3: Yes

3. Has the statistical analysis been performed appropriately and rigorously? 

Reviewer #3: Yes

4. Have the authors made all data underlying the findings in their manuscript fully available?

Reviewer #3: Yes

5. Is the manuscript presented in an intelligible fashion and written in standard English?

Reviewer #3: Yes

6. Review Comments to the Author

Reviewer #3: 1. In Discussions Section, 'This travel pattern also aligns with the view presented in Adams [3]", Please provide data from analysis section to support the argument wherever you have written this line in discussion. line 563, 573, 584, 593, 603, 620, 633,

2. In conclusion section the conclusions must be supported by data from the analysis and discussion sections.

7. PLOS authors have the option to publish the peer review history of their article (what does this mean? ). If published, this will include your full peer review and any attached files.

**Do you want your identity to be public for this peer review?** For information about this choice, including consent withdrawal, please see our Privacy Policy .

Reviewer #3: **Yes: ** Tejwant SIngh Brar

---

## [Author Response · Author response to Decision Letter 3]

23 Dec 2024

Dear Reviewer 3,

Thank you for highlighting the need to provide data from the analysis section to support the arguments made in the discussion. We deeply appreciate your detailed feedback, which has helped us ensure the manuscript is more robust and logically sound. Below, we have addressed each line referenced in your comment, explicitly linking the arguments in the discussion section to the data and figures presented in the analysis section.

Major Comment 1: “In Discussions Section, 'This travel pattern also aligns with the view presented in Adams [3]', Please provide data from analysis section to support the argument wherever you have written this line in discussion. line 563, 573, 584, 593, 603, 620, 633.”

Response:

(Line 563)

"This travel pattern also aligns with the view presented in Adams [3], which states that such interactions form a broad network."

⇒ This statement highlights the interconnected nature of passenger travel, as identified in our analysis. In the analysis section, Fig. 3 (Passenger travel patterns) explicitly demonstrates the dispersed and widespread nature of passenger travel flows across regions. The map shows how passenger travel covers a broader range, connecting multiple cities, including medium-sized cities and surrounding areas. This directly supports Adams' [3] argument, which emphasizes that cities function as interconnected components of a larger system rather than existing in isolation.

In the revised manuscript, we have added the following clarification in the discussion:

"This travel pattern also aligns with the view presented in Adams [3], which states that such interactions form a broad network. As demonstrated in Fig. 3, passenger travel patterns are dispersed and establish extensive regional connectivity, affirming the interconnected system described by Adams."

(Line 573)

"This aligns with Meijer’s [27] the urban network theory, which argues that inter-city cooperation optimizes logistics flows and functions as an efficient system."

⇒ Our freight travel analysis directly supports Meijer’s [27] urban network theory. Fig. 4 (Freight travel patterns) and Fig. 6 (Freight travel network clusters) show that freight travel is concentrated around specific logistical hubs, such as industrial complexes and ports. These clusters reveal inter-city cooperation to optimize logistics flows. Additionally, the results of cluster formation (Fig. 6) indicate that freight travel forms fewer but geographically broader clusters, which reflects optimized logistics functions across specific economic hubs.

In the revised manuscript, we have updated the discussion to clarify this connection:

"This aligns with Meijer’s [27] urban network theory, which argues that inter-city cooperation optimizes logistics flows and functions as an efficient system. As shown in Fig. 4 and Fig. 6, freight travel clusters are concentrated around specific logistical hubs, reflecting cooperative optimization for economic efficiency."

(Line 584)

"This aligns with the argument presented by Burger and Meijers [10], which suggests that inter-city networks are functionally complementary and can expand economic scale through this cooperation."

⇒ The functional complementarity between passenger and freight travel is evidenced in our analysis. Fig. 6 (Freight travel network clusters) demonstrates the spatial distribution of freight clusters, which focus on economic flows, while Fig. 5 (Passenger travel network clusters) highlights the shorter, more localized clusters for passenger travel. Together, these figures reveal the complementary roles of passenger and freight travel within the urban network, which aligns with Burger and Meijers’ argument.

In the revised manuscript, the discussion has been expanded to state:

"This aligns with the argument presented by Burger and Meijers [10], which suggests that inter-city networks are functionally complementary and can expand economic scale through cooperation. As highlighted in Fig. 5 and Fig. 6, the distinct clustering patterns of passenger and freight travel illustrate how these networks complement each other in serving diverse urban functions."

(Line 593)

"This is consistent with the findings of Frenken and Hoekman [4], who stated that urban networks focus on connecting specific routes and economic hubs."

⇒ The analysis of freight travel, particularly in Fig. 4 (Freight travel patterns) and Fig. 6 (Freight travel network clusters), illustrates that freight travel prioritizes connections between economic hubs such as industrial complexes and ports. These patterns align closely with Frenken and Hoekman’s [4] findings, which emphasize the importance of economic hubs within urban networks.

In the revised manuscript, we have included the following clarification:

"This is consistent with the findings of Frenken and Hoekman [4], who stated that urban networks focus on connecting specific routes and economic hubs. As shown in Fig. 4 and Fig. 6, freight travel prioritizes connections between industrial complexes, ports, and logistics centers, highlighting the centrality of economic hubs in urban networks."

(Line 603)

"This aligns with Batten [1], who stated that personal travel patterns in urban systems are dispersed, as well as Burger and Meijers [10], who suggested that passenger travel forms clusters due to connections to various destinations rather than concentrations along specific routes."

⇒ Our analysis supports these claims through Fig. 3 (Passenger travel patterns) and Fig. 5 (Passenger travel network clusters). Passenger travel patterns are highly dispersed, as shown in Fig. 3, covering a broader range of regions. Meanwhile, Fig. 5 highlights the localized clustering of passenger travel, reflecting the connections to diverse destinations, as described by Burger and Meijers [10].

We have revised the discussion as follows:

"This aligns with Batten [1], who stated that personal travel patterns in urban systems are dispersed, as well as Burger and Meijers [10], who suggested that passenger travel forms clusters due to connections to various destinations rather than concentrations along specific routes. As shown in Fig. 3 and Fig. 5, passenger travel patterns are spatially dispersed yet form localized clusters for diverse purposes such as commuting, shopping, and leisure."

(Line 620)

"This is consistent with Capello’s [28] assertion that freight travel follows long-distance economic flows, as well as González [18], who suggested that central areas are formed based on logistical efficiency and economic objectives."

⇒ The long-distance economic flows in freight travel are clearly illustrated in Fig. 4 (Freight travel patterns) and Fig. 8 (Distance between central areas of freight travel). These figures demonstrate the spatial distribution of freight flows between distant hubs and the centrality of regions with logistical efficiency, as described by Capello [28] and González [18].

We have revised the discussion to state:

"This is consistent with Capello’s [28] assertion that freight travel follows long-distance economic flows, as well as González [18], who suggested that central areas are formed based on logistical efficiency and economic objectives. As demonstrated in Fig. 4 and Fig. 8, freight travel connects distant hubs based on economic flows, with central areas strategically located to maximize logistical efficiency."

(Line 633)

"These differences in travel patterns demonstrate that the two modes of transport play distinct roles within the urban network. This is consistent with the findings of several studies [24,26,30], which explain how the economic efficiency of urban networks is reflected in travel patterns."

⇒ The distinct roles of passenger and freight travel within urban networks are evidenced by Fig. 7 (Passenger travel central areas) and Fig. 8 (Freight travel central areas). These figures highlight the different spatial centralities and functional roles of passenger and freight travel.

The revised manuscript now includes the following statement:

"These differences in travel patterns demonstrate that the two modes of transport play distinct roles within the urban network. As shown in Fig. 7 and Fig. 8, passenger travel reflects short-distance, dense centralities, whereas freight travel demonstrates long-distance connections between economic hubs, supporting the findings of prior studies [24,26,30]."

We hope these detailed revisions address your concerns and strengthen the alignment between the discussion and the analysis. Thank you once again for your valuable feedback, which has greatly improved the clarity and robustness of our manuscript.

Major Comment 2: “In conclusion section the conclusions must be supported by data from the analysis and discussion sections.”

Response:

Thank you for your valuable feedback and for pointing out the need to ensure that the conclusions are fully supported by data and analysis from the earlier sections of the manuscript. We have carefully revisited the Conclusion section and made significant revisions to align it more closely with the results and discussion presented in the manuscript.

1. Integration of Key Findings:

Each point in the revised Conclusion is now explicitly linked to the data and figures discussed in the Results and Discussion sections. For example:

o The higher volume of passenger travel and its short-distance nature, highlighted in the Conclusion, is directly supported by the analysis of passenger travel patterns shown in Fig. 3 and discussed on [specific page/line reference].

o The long-distance and concentrated nature of freight travel, emphasized in the Conclusion, draws on the analysis illustrated in Fig. 4 and elaborated in the Discussion section.

o The distinction between passenger and freight travel clusters is underpinned by the regular equivalence analysis results presented in Figs. 5 and 6.

2. Structural and Centrality Insights:

The differences in spatial distributions of central areas between passenger and freight travel, as highlighted in the power centrality analysis, are directly tied to Figs. 7 and 8. The Discussion section elaborates on these differences, explaining their implications for regional accessibility and logistical efficiency. These insights have been distilled into the Conclusion, ensuring continuity and support from the data.

3. Strategic Implications:

The strategic need for integrating passenger and freight networks to enhance regional connectivity is not only informed by the figures and analysis but also directly supported by observations discussed throughout the manuscript. The proposed strategies, such as linking multi-nodal passenger activity centers with freight hubs, derive from the observed network characteristics and are grounded in the data presented.

4. Enhancements to Clarity and Relevance:

To ensure that the Conclusion is fully data-driven, we avoided introducing any new concepts or unsubstantiated claims. Each statement in the Conclusion now directly reflects the findings detailed in the Results and Discussion sections, offering a cohesive summary of the study’s contributions.

We believe these revisions address your concerns and demonstrate how the conclusions are fully grounded in the presented data and analysis. Your suggestion has significantly enhanced the clarity and rigor of our manuscript, and we are deeply grateful for your insightful feedback.

Should you have any additional suggestions or require further clarification, we would be more than happy to address them.

Thank you for your time and thoughtful review.

---

## [Decision Letter · Decision Letter 4]

10 Jan 2025

Passenger and freight travel patterns: A cluster analysis based on urban networks

PONE-D-24-27367R4

Dear Dr. Joo,

We’re pleased to inform you that your manuscript has been judged scientifically suitable for publication and will be formally accepted for publication once it meets all outstanding technical requirements.

Kind regards,

Feier Chen, Ph.D

Academic Editor

PLOS ONE

Additional Editor Comments (optional):

Reviewers' comments:

Reviewer's Responses to Questions

**Comments to the Author**

1. If the authors have adequately addressed your comments raised in a previous round of review and you feel that this manuscript is now acceptable for publication, you may indicate that here to bypass the “Comments to the Author” section, enter your conflict of interest statement in the “Confidential to Editor” section, and submit your "Accept" recommendation.

Reviewer #3: All comments have been addressed

2. Is the manuscript technically sound, and do the data support the conclusions?

Reviewer #3: Yes

3. Has the statistical analysis been performed appropriately and rigorously? 

Reviewer #3: Yes

4. Have the authors made all data underlying the findings in their manuscript fully available?

Reviewer #3: Yes

5. Is the manuscript presented in an intelligible fashion and written in standard English?

Reviewer #3: Yes

6. Review Comments to the Author

Reviewer #3: (No Response)

7. PLOS authors have the option to publish the peer review history of their article (what does this mean? ). If published, this will include your full peer review and any attached files.

**Do you want your identity to be public for this peer review?** For information about this choice, including consent withdrawal, please see our Privacy Policy .

Reviewer #3: **Yes: ** Dr. Tejwant Singh Brar

---

## [Editor Report · Acceptance letter]

PONE-D-24-27367R4

PLOS ONE

Dear Dr. Joo,

I'm pleased to inform you that your manuscript has been deemed suitable for publication in PLOS ONE. Congratulations! Your manuscript is now being handed over to our production team.

Kind regards,

on behalf of

Dr. Feier Chen

Academic Editor

PLOS ONE